# Just How Flexible are Neural Networks in Practice?

## Abstract

Although overparameterization theory suggests that neural networks can fit any dataset with up to as many samples as they have parameters, practical limitations often prevent them from reaching this capacity. In this study, we empirically investigate the practical flexibility of neural networks and uncover several surprising findings. Firstly, we observe that standard optimizers, such as stochastic gradient descent (SGD), often converge to solutions that fit significantly fewer samples than the model's parameter count, highlighting a gap between theoretical and practical capacity. Secondly, we find that convolutional neural networks (CNNs) are substantially more parameter-efficient than multi-layer perceptrons (MLPs) and Vision Transformers (ViTs), even when trained on randomly labeled data, emphasizing the role of architectural inductive biases. Thirdly, we demonstrate that the difference in a network's ability to fit correctly labeled data versus incorrectly labeled data is a strong predictor of generalization performance, offering a novel metric for predicting generalization. Lastly, we show that stochastic training methods like SGD enable networks to fit more data than full-batch gradient descent, suggesting that stochasticity enhances flexibility beyond regularization effects. These findings highlight the importance of understanding practical capacity limits and their implications for model generalization, providing new insights into neural network training and architectural design.

## 1 Introduction

Understanding the capacity and flexibility of neural networks is fundamental to advancing deep learning research and applications. It is widely believed that neural networks can fit any training set containing at most as many samples as they have parameters. This belief stems from theoretical results in universal approximation (Hornik et al., 1989) and empirical observations where large neural networks can perfectly fit their training data, even with random labels (Zhang et al., 2016).

However, in practice, the solutions found by neural networks are heavily influenced by training procedures, including the choice of optimizer, regularization techniques, and architectural design. These factors shape the loss landscape and determine which minima are accessible during training. Consequently, neural networks may not fully utilize their theoretical capacity to fit data, and their practical flexibility can be significantly limited.

Moreover, the specific parameterization inherent in a neural network's architecture can affect its ability to fit data. For example, convolutional neural networks (CNNs) exhibit inductive biases such as locality and translation invariance, which impact their performance on image data. Understanding how these architectural choices influence the practical capacity of neural networks is crucial for designing efficient and effective models.

To systematically study the practical flexibility of neural networks, we adopt the *Effective Model Complexity* (EMC) metric (Nakkiran et al., 2021), which estimates the largest sample size that a model can perfectly fit using realistic training procedures. Unlike theoretical capacity measures, EMC accounts for the effects of optimization algorithms, regularization techniques, and data properties on a model's ability to fit data.

In this paper, we conduct extensive experiments to analyze how data properties, model architectures, and training procedures influence EMC. Our key findings are:

- **Limited Practical Capacity:** Standard training procedures often lead to solutions where neural networks can only fit datasets containing significantly fewer samples than the number of model parameters.
- **Architecture Matters:** CNNs are more parameter-efficient than MLPs and ViTs, even when trained on randomly labeled data, indicating that architectural biases play a critical role in practical capacity.
- **Optimizer Influence:** Stochastic Gradient Descent (SGD) enables models to fit more training data compared to full-batch gradient descent, suggesting that stochasticity may enhance the capacity to fit data rather than acting solely as a regularizer.
- **Predicting Generalization:** The difference in EMC when fitting correctly labeled versus incorrectly labeled data correlates strongly with generalization performance, providing a practical tool for predicting a model's ability to generalize.
- **Activation Functions:** ReLU activation functions improve a model's ability to fit data beyond their role in addressing vanishing and exploding gradients in deep networks.

Our findings provide new insights into the practical limitations and capabilities of neural networks, highlighting the importance of training dynamics and architectural choices in determining a model's capacity to fit data and generalize effectively.

## 2 RELATED WORK

**Approximation Theory.** Early deep learning theory focused on the function approximation capabilities of neural networks. The universal approximation theorem established that feedforward networks with a single hidden layer can approximate any continuous function on compact subsets of $\mathbb{R}^n$, given sufficient width (Hornik et al., 1989). Subsequent studies provided upper bounds on the number of parameters required for specific function classes (Barron, 1993; Mhaskar & Poggio, 2016). These approximation theories typically address arbitrary compact sets or data on well-behaved manifolds (Shaham et al., 2018) and often apply to shallow networks, limiting their practical applicability. In contrast, our work empirically measures neural network flexibility on real datasets, considering various architectures and training procedures to capture factors that directly impact practical capacity.

**Overparameterized Neural Networks and Generalization.** Traditional generalization theories based on VC-dimension and Rademacher complexity suggest that models with low capacity should generalize well (Vapnik, 1991; Bartlett & Mendelson, 2002). However, these theories do not explain why highly overparameterized neural networks can generalize effectively even when they can fit random labels (Zhang et al., 2016). Recent advancements in PAC-Bayes generalization theory have provided frameworks to understand how overparameterized models can still generalize well by assigning disproportionate prior mass to parameter vectors that fit the training data (Dziugaite & Roy, 2017; Lotfi et al., 2022; 2023). Empirical studies have further demonstrated that inductive biases and overparameterization can enhance generalization (Huang et al., 2019; Chiang et al., 2022; Maddox et al., 2020). Specifically, Nakkiran et al. (2021) explored the data-fitting capacity of neural networks to understand the double-descent phenomenon, showing that overparameterization interacts with data properties and training dynamics to influence generalization.

In contrast, our study investigates the factors that influence the capacity of neural networks to fit data by adopting the metric *Effective Model Complexity* (EMC) (Nakkiran et al., 2021). We empirically measure the largest sample size that a model can perfectly fit under realistic training conditions and investigate how EMC predicts generalization performance. This approach bridges theoretical insights with empirical observations, providing a comprehensive understanding of neural network flexibility and its implications for generalization.

## 3 MODEL CAPACITY AND EMPIRICAL COMPLEXITY

**Quantifying Capacity.** Determining how many samples a model can perfectly fit is straightforward in the case of linear models, where capacity directly corresponds to the number of parameters. However, neural networks introduce complexity that goes beyond parameter counting. Our goal is to

define a practical metric that captures a neural network's capacity to fit data under real-world training conditions. This metric must meet three key criteria: (1) it should reflect the capacity to fit real-world data, accounting for the influence of optimizers, regularization, and data augmentation; (2) it must be sensitive to the characteristics of the training data, including the nature of the inputs and labels; and (3) it should be computationally feasible to apply across diverse architectures and datasets.

To address these criteria, we employ the *Effective Model Complexity* (EMC) metric (Nakkiran et al., 2021), which quantifies the largest sample size a model can perfectly fit using realistic training routines. Unlike theoretical capacity measures that are architecture-specific or focus on idealized conditions, EMC captures the impact of practical training dynamics, including optimization procedures, regularization strategies, and data properties.

**Computing EMC.**   Calculating EMC involves an iterative approach for each network size. Initially, we train the model on a small number of samples. If it achieves 100% training accuracy, we re-initialize and train on a larger set of randomly chosen samples. We iteratively perform this process, incrementally increasing the sample size each time until the model no longer fits all training samples perfectly. The largest sample size where the model still achieves perfect fitting is taken as the network's EMC. Importantly, we ensure that the initialization and data subsets are independent at each iteration to maintain an unbiased capacity evaluation. Furthermore, we also tried performing all analyses instead with a relaxed requirement that the network fit $98\%$ of its training data, which did not significantly affect results.

While it is possible to artificially prevent models from fitting their training set by under-training, confounding any study of capacity, we ensure that all training runs reach a minimum of the loss function by imposing three conditions: (1) the norm of the gradients across all samples must fall below a predefined threshold; (2) the training loss should stabilize; (3) we check for the absence of negative eigenvalues in the loss Hessian to confirm that the model has reached a minimum rather than a saddle point.

**The differences between capacity, flexibility, expressiveness, and complexity.**   These terms are used in numerous ways, sometimes interchangeably and sometimes distinctly. For example, Rademacher complexity and VC-dimension are notions of complexity typically associated with flexibility, whereas the PAC-Bayes notion of complexity is information-theoretic and instead measures compression. Expressiveness can describe the breadth of an entire hypothesis class, that is, all the functions a model can express across all possible parameter settings. Approximation theories measure the expressiveness of a hypothesis class by the existence of elements of this class, which are well-approximate functions of a specified type. We will abstain from using the terms "expressiveness" and "complexity" when describing EMC to avoid confusion, and we will use "capacity" and "flexibility" when referring to a model's ability to fit data in practice.

**Factors Influencing EMC.**   Unlike VC-dimension or expressiveness concepts in approximation theories, EMC depends not only on the hypothesis class but on every aspect of neural network training, from optimizers and regularizers to the specific parameterization induced by the model's architecture. Choices in architectural design and training algorithms influence the loss surface geometry, thereby affecting the accessibility of certain solutions.

## 4   EXPERIMENTAL SETUP

We perform a comprehensive dissection of the factors influencing neural network flexibility. To this end, we consider a variety of datasets, architectures, and optimizers.

### 4.1   DATASETS

We perform experiments on a range of datasets, including image datasets like MNIST (Deng, 2012), CIFAR-10, CIFAR-100 (Krizhevsky et al., 2009), and ImageNet (Deng et al., 2009), as well as tabular datasets like Forest Cover Type (Blackard & Dean, 1999), Adult Income (Becker & Kohavi, 1996), and the Credit dataset (Kaggle, 2021). Due to the small size of these datasets, we also use larger synthetic datasets generated using the Efficient Diffusion Training via Min-SNR Weighting

Strategy (Hang et al., 2023), yielding diverse ImageNet-quality samples at a resolution of $128 \times 128$. Specifically, we create ImageNet-20MS, containing 20 million samples across ten classes. Unless otherwise specified, the main text describes results on ImageNet-20MS, while the appendix contains results on additional datasets. We omit data augmentations to avoid confounding effects.

### 4.2 MODELS

We evaluate the flexibility of diverse architectures, including MLPs, CNNs such as ResNet (He et al., 2016) and EfficientNet (Tan & Le, 2019), and ViTs (Dosovitskiy et al., 2020). We systematically adjust the width and depth of these architectures. For MLPs, we either increase the width by adding neurons per layer while keeping the number of layers constant or increase the depth by adding more layers while keeping the number of neurons per layer constant. For naive CNNs, we employ multiple convolutional layers followed by a constant-sized fully connected layer, varying either the number of filters per layer or the total number of layers. For ResNets, we scale either the number of filters or blocks (depth). In ViTs, we scale the number of encoder blocks (depth), the dimensionality of patch embeddings, and self-attention (width). By default, we scale the width unless otherwise stated.

### 4.3 OPTIMIZERS

We employ several optimizers, including SGD, Adam (Kingma & Ba, 2015), AdamW (Loshchilov & Hutter, 2018), full-batch Gradient Descent (GD), and the second-order Shampoo optimizer (Anil et al., 2021). These choices let us examine how features like stochasticity and preconditioning influence the minima. To ensure effective optimization across datasets and model sizes, we carefully tune the learning rate and batch size for each setup, omitting weight decay in all cases. Further details about our hyperparameter tuning are provided in Appendix B. By default, we use SGD.

## 5 THE EFFECT OF THE DATA ON EMC

In this section, we explore how data properties shape neural network flexibility and how this behavior can predict generalization.

### 5.1 ANALYSIS OF DIVERSE DATASETS

We initiate our analysis by measuring the EMC of neural networks across various datasets and modalities. We scale a 2-layer MLP by modifying the width of the hidden layers and a CNN by modifying the number of layers and channels, and we train models on a range of image classification (MNIST, CIFAR-10, CIFAR-100 and ImageNet) and tabular (Forest Cover Type, Income, and Credit) datasets. The results reveal significant disparities in the EMC of networks trained on different data types (see Figure 1 (left)). For instance, networks trained on tabular datasets exhibit higher capacity. Among image classification datasets, we observe a strong correlation between test accuracies and capacity. Notably, MNIST (where models achieve more than 99% test accuracy) yields the highest EMC, whereas ImageNet shows the lowest, pointing to the relationship between generalization and data-fitting capability.

Considering the variety of datasets and network architectures and the myriad differences in their EMC, the following sections will explore the underlying causes of these variations. Our goal is to identify the distinct factors in the data and architectures contributing to these observed network flexibility differences.

### 5.2 THE ROLE OF INPUTS AND LABELS

We next analyze how architectural inductive biases and factors like spatial structure influence a model's ability to fit data. By altering both inputs and labels, we measure the effects on EMC. Specifically, we adjust the width of MLPs and 2-layer CNNs by varying neurons or filters and train these models on the ImageNet-20MS dataset under four scenarios: semantic labels, random labels, random inputs, and inputs with fixed random permutation.

In the random labels case, we keep the original inputs but assign random class labels, disrupting the semantic structure. For random inputs, we replace images with Gaussian noise, removing meaningful

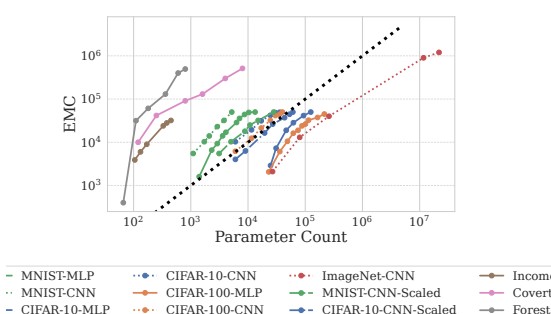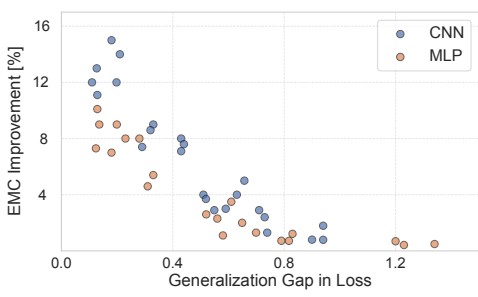

Figure 1: **Left: easier tasks tend to have higher EMC.** EMC across datasets and data modalities. The tabular data sets (Forest, Income, CoverType), which are easier to learn, have the highest EMC compared to vision datasets. The dashed black line is the diagonal. ImageNet is the hardest dataset to learn. **Right: The difference in EDC on the original and random labels predicts generalization.** EMC improvement as a function of the parameter count for CIFAR-100.

spatial information. For permuted inputs, we apply a fixed random permutation to all images, breaking the spatial structure while preserving the input dimensions.

### 5.2.1 THE BOUNDARY BETWEEN OVERPARAMETERIZATION AND UNDERPARAMETERIZATION

Linear regression models can fit at least as many samples as they have parameters, regardless of whether the labels are naturally occurring or random. The boundary between where a model has too few parameters to fit its data and where it has extra degrees of freedom is clear for linear regression. Naturally occurring labels present a more complicated scenario; for instance, if the data's labels are a linear function of the inputs, the model can fit infinitely many samples. In Figure 2, assigning random labels instead of real ones allows us to explore an analogous notion of the boundary between over- and under-parameterization, but in the context of neural networks. We see here that the networks fit significantly fewer samples when assigned random labels compared to the original labels, indicating that neural networks are less parameter efficient than linear models in this setting. Similar to linear models, the amount of data they can fit scales linearly with parameter count.

### 5.2.2 THE EFFECT OF HIGH-DIMENSIONAL DATA

Linear models exhibit increased capacity when adding more features, primarily because their parameter count directly scales with the feature count. However, the dynamics shift when examining CNNs. In our setup, we avoid adding parameters as the data dimensionality increases by employing average pooling prior to the classification head, a standard technique for CNNs. We investigated the EMC using ImageNet-20MS, systematically resizing the input images to vary their spatial dimensions from $16 \times 16$ to $256 \times 256$.

In contrast to linear models, we find (Figure 17 in the Appendix) that CNNs, which do not benefit from additional parameters as the input dimensionality increases, can fit more semantically labeled data in lower spatial dimensions. This trend underscores a broader narrative in neural networks: CNNs, despite their intricate architectures and capacity for complex pattern recognition, tend to align better with data of lower intrinsic dimension. This observation resonates with the findings of Pope et al. (Pope et al., 2020), who found that CNNs generally show enhanced generalization capabilities with data of lower intrinsic dimensionality.

To further investigate the impact of input dimensions on EMC, we conducted an additional experiment by scaling the CIFAR-10 and MNIST datasets to the ImageNet input size of $256 \times 256$ pixels (Figure 1 - left). Remarkably, these scaled datasets maintained higher EMC values compared to ImageNet itself, despite having identical input dimensions. This finding suggests that intrinsic dataset complexity plays a significant role in influencing fitting capacity, beyond merely the size of the input.

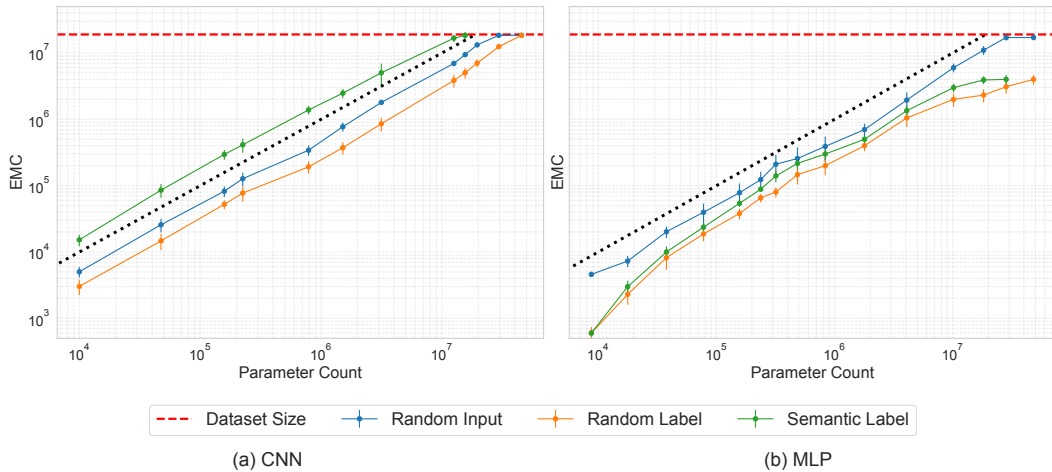

Figure 2: **CNNs fit more semantically labeled samples than they have parameters due to their superior image classification inductive bias, whereas MLPs cannot.** EMC as a function of the number of parameters for semantic labels vs. random input and labels for MLPs **(a)** and CNNs **(b)**. Experiments performed on ImageNet-20MS. The error bars represent one standard error over 5 trials.

This experiment underscores that factors such as the inherent complexity and variability of the dataset are crucial determinants of a model's capacity to fit data, independently of input dimensionality. It highlights that simply increasing input size does not linearly translate to increased fitting capacity if the intrinsic complexity of the dataset remains high.

### 5.2.3 THE EFFECT OF THE NUMBER OF CLASSES

We examined how the number of classes influences EMC through experiments on the CIFAR-100 and ImageNet-20MS datasets.

**Experiment 1: Merging Classes in CIFAR-100** By randomly merging classes in CIFAR-100, we reduced the number of classes while keeping the dataset size constant. Using 2-layer CNNs with varying parameters, we observed (Figure 3a - left) that increasing the number of classes made it harder to fit data with semantic labels, as the model must encode more complex information. Conversely, with randomly labeled data, more classes made fitting easier since the model's inductive bias towards correct labeling is less constrained.

**Experiment 2: Varying Class Numbers in ImageNet-20MS** To isolate the effect of class count, we increased the number of classes in ImageNet-20MS without altering intra-class variance or total sample size. The results (Figure 23 in the Appendix) confirmed that EMC decreases with more classes even when within-class diversity is controlled, reinforcing that the number of classes independently affects the difficulty of fitting semantic labels.

**Experiment 3: Binary Classification Across Multiple Datasets** To compare different datasets while controlling the number of classes, we converted several datasets into binary classification problems. By reducing the classification task to binary, we eliminate the variability introduced by having multiple classes, focusing solely on how the effect of the input distribution. Our results (Figure 22 in the Appendix) indicate that even when the number of classes is controlled, different datasets exhibit varying EMC levels. This suggests that factors like intrinsic data complexity and distribution characteristics significantly impact a model's capacity to fit data beyond just the number of classes.

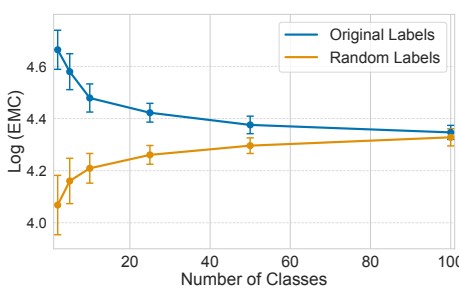 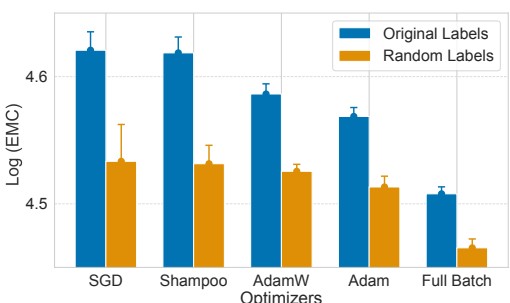

(a) **More classes make fitting data harder with se-mantic labels but easier with random ones.**

(b) **SGD and Shampoo enable fitting more original labels but random ones.**

Figure 3: **The effect of the number of labels and optimizers on capacity.** Average logarithm of EMC across different model sizes of CNNs on CIFAR-100 for original and random labels varying numbers of classes **(a)** and for different optimizers **(b)**. Error bars are standard error over 5 trials.

# 6 EMC as a Predictor of Generalization Performance

Neural networks tend to fit semantically coherent labels more readily than random ones, reflecting their inductive biases. This preference, shown in Figure 1 (right), suggests that a network's ability to fit semantic labels correlates with its generalization performance. This enables architectures like CNNs to fit more samples than their parameter count might suggest, blurring traditional boundaries between under- and over-parameterization.

This observation connects two perspectives on generalization. Traditional wisdom posits that high-capacity models overfit and fail to generalize-a notion reflected in early generalization bounds, which are vacuous for neural networks (Vapnik, 1991; Bartlett & Mendelson, 2002). In contrast, PAC-Bayes theory proposes that a model's flexibility does not impede generalization if it assigns more prior mass to true labels than to random ones (Dziugaite & Roy, 2017). Our empirical findings relate these theories by showing a strong relationship between a model's increased ability to fit correct labels over random ones and its generalization performance.

To quantify this, we computed the EMC for various CNN and MLP configurations on both correctly and randomly labeled data. We gauged the practical capacity to fit natural label distributions by easuring the percentage increase in EMC for semantic labels. The significant inverse correlation between this metric and the generalization gap (Pearson correlation coefficients of $-0.9281$ for CNNs and $-0.869$ for MLPs) confirms theoretical underpinnings and highlights practical implications (Figure 1 - right).

To further assess EMC's predictive power, we compared it with other generalization predictors, including relative flatness (Petzka et al., 2021), $L_2$ weight norm, trace of Hessian (Liu et al., 2022), Fisher-Rao norm (Liang et al., 2019), PAC-Bayes criteria (Jiang* et al., 2020), and an information-theoretic bound from Kawaguchi et al. (2023). As shown in Table 1, our EMC-based measure achieves the highest correlation with generalization, firmly establishing its superiority as a generalization predictor.

| Measure | Spearman | Pearson |
|---|---|---|
| Weight Norm | 0.758 | 0.615 |
| Trace of Hessian | 0.784 | 0.758 |
| Fisher-Rao Norm | 0.622 | 0.237 |
| PAC-Bayes Criteria | 0.823 | 0.853 |
| Relative Flatness | 0.837 | 0.799 |
| Information Bottleneck | 0.851 | 0.839 |
| **EMC (Ours)** | **0.868** | **0.853** |

Table 1: **EMC outperforms other generalization metrics.** Spearman and Pearson correlations with generalization error on CIFAR-10.

# 7 THE EFFECT OF MODEL ARCHITECTURE ON EMC

Having analyzed the influence of data on flexibility, we now focus on the impact of architecture. We examine how different architectural properties—such as MLPs, CNNs, transformers, activation functions, and scaling strategies—contribute to flexibility.

## 7.1 ARCHITECTURAL STYLE AND PARAMETER EFFICIENCY

To address the debate on the efficiency and generalization of CNNs versus Vision Transformers (ViTs) (d'Ascoli et al., 2021; Patro & Agneeswaran, 2023; Maurício et al., 2023; Goldblum et al., 2023), we evaluated three architectures: MLPs, CNNs, and ViTs.

Our findings (Figure 4b) reveal that CNNs, characterized by hard-coded inductive biases like locality and translation equivariance, consistently outperform ViTs and MLPs in EMC (see Figure 18 in the Appendix for detailed scaling laws). This superiority holds across all model sizes on semantically labeled data. As analyzed in the previous section, this trend could be misconstrued as a result of better generalization over ViTs and MLPs.

To test, we examine the networks' flexibility on randomized data (Figure 18 in the Appendix). CNNs, which rely on spatial structure, fit fewer samples when spatial structure is destroyed via permutation. MLPs, lacking this preference, show unchanged flexibility. Replacing inputs with Gaussian noise increases both architectures' capacity, possibly because high-dimensional noisy data is easier to separate. Notably, CNNs fit far more samples with semantic labels than with random inputs, while MLPs show the opposite trend, underscoring CNNs' superior generalization in image classification.

Despite random data affecting architectures differently, the hierarchy of parameter efficiency remains the same. This consistency suggests that CNNs' superior parameter efficiency is inherent to their architectural design, not merely a result of better generalization. This aligns with approximation theory, which posits CNNs as more parameter-efficient than MLPs (Bao et al., 2014).

## 7.2 STRATEGIES FOR SCALING NETWORK SIZE

The debate on how to best scale width and depth in neural networks is ongoing. We now focus on how different scaling strategies affect a network's ability to fit data (Figure 4a and Figure 5 in the Appendix). For ResNets, we consider increasing width (number of filters), increasing depth, or scaling both using EfficientNet (Tan & Le, 2019) and ResNet-RS (Bello et al., 2021) scaling laws. EEfficientNet balances the scaling of depth, width, and resolution; ResNet-RS adapts the scaling based on model size, training time, and dataset size. For ViTs, we use the SViT (Zhai et al., 2022), SoViT (Alabdulmohsin et al., 2023), and also scale depth and width separately.

Our analysis shows that these scaling laws, although not originally aimed at optimizing capacity, perform well in improving EMC. Consistent with theoretical analyses (Eldan & Shamir, 2016), we find that scaling depth is more parameter-efficient than scaling width. These findings hold even on randomly labeled data, suggesting the results are not due to generalization effects.

## 7.3 ACTIVATION FUNCTIONS

Nonlinear activation functions are essential for neural network capacity; without them, networks reduce to linear models. We investigate how different activation functions affect EMC, contrasting nonlinear activations with linear models.

Our findings (Figure 16 in the Appendix) indicate that ReLU activations significantly enhance capacity. Originally introduced to mitigate vanishing and exploding gradients, ReLU also improves the network's ability to fit data, likely by enhancing generalization. In contrast, tanh activation, although nonlinear, does not yield similar capacity improvements, even though we can still find minima with it. This suggests that ReLU's contribution to increased capacity is not solely due to easier optimization.

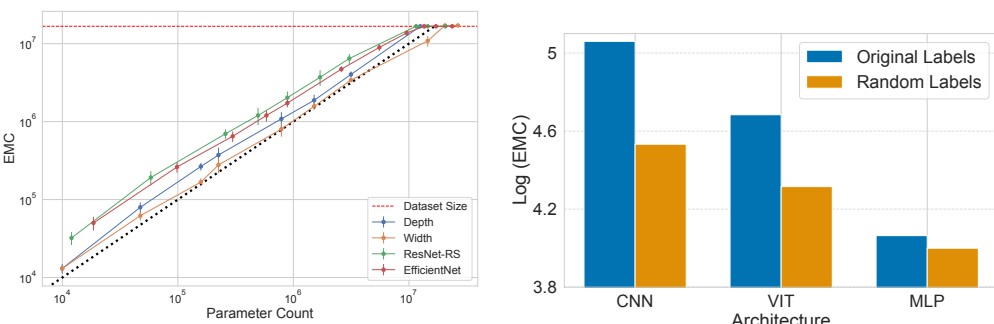

(a) **ResNet-RS is the most efficient among scaling strategies we test.**

(b) **CNNs are far more parameter-efficient, even on randomly labeled data.**

Figure 4: **The effect of the scaling strategy and the architecture on the EMC** . **(a)** Scaling laws for the EMC as a function of parameters counts for CNN. **(b)** Average logarithm of EMC across parameter counts for different architectures using original and random labels. On ImageNet-20MS. Error bars represent one standard error over 5 trials.

## 8 THE ROLE OF OPTIMIZATION IN FITTING DATA

Optimization techniques and regularization strategies are crucial in neural network training, influencing convergence rates and the nature of solutions obtained. We explore how different optimization algorithms and regularization methods affect a network's capacity to fit data, as measured by EMC.

### 8.1 COMPARING OPTIMIZERS

We evaluated various optimizers, including SGD, full-batch GD, Adam (Kingma & Ba, 2015), AdamW (Loshchilov & Hutter, 2018), and Shampoo (Gupta et al., 2018), to assess their impact.

While previous work suggests that SGD has a strong flatness-seeking regularization effect leading to better generalization (Geiping et al., 2021), our findings reveal a more nuanced picture. As shown in Figure 3b, SGD enables models to fit more data than full-batch GD, achieving EMC levels comparable to the sophisticated Shampoo optimizer. This suggests that the stochasticity inherent in SGD helps the model discover minima with a higher capacity to fit data.

However, with randomly labeled data, the higher EMC of SGD and Shampoo diminished, indicating that their enhanced capacity is connected to their superior generalization on the original tasks.

### 8.2 EFFECT OF REGULARIZATION TECHNIQUES

Classical machine learning often employs regularizers designed to reduce model capacity and prevent overfitting. For example, ridge regression applies a penalty on the parameter norm to improve the performance of overparameterized linear models (Hoerl & Kennard, 1970). We investigated whether these regularizers impact a model's capacity to fit data.

We evaluated the EMC of a CNN trained on ImageNet-20MS using different regularization methods, including Sharpness-Aware Minimization (SAM) (Foret et al., 2020), weight decay, and label smoothing (Müller et al., 2019) (Figure 15 in the Appendix). We found that weight decay and label smoothing reduced the EMC, indicating a decreased capacity to fit data. In contrast, SAM improved generalization without diminishing EMC, even on randomly labeled data. This suggests that SAM facilitates the discovery of minima that generalize better without sacrificing the ability to fit large amounts of data. Notably, label smoothing modifies the loss function itself, potentially hindering the model from finding minima of the original objective, while SAM preserves the original loss landscape.

## 8.3 SGD Solutions' Flatness and EMC

We further investigated the relationship between the flatness of solutions found by SGD and their EMC. Our findings, illustrated in Figure 21 in the Appendix, show that SGD consistently finds flatter solutions with higher EMC than full-batch GD. This indicates that achieving a higher capacity to fit data does not come at the expense of flatness. Our results challenge the conventional belief that flatter minima are associated with lower capacity, suggesting that flatness and high EMC can coexist.

## 9 Reparameterization for Increased Parameter Efficiency

We observed that neural networks often require more parameters than the number of samples they can fit, indicating inefficiency in parameter utilization. To address this, we explored two reparameterization strategies aimed at increasing parameter efficiency.

First, we employed *subspace training* (Lotfi et al., 2022), projecting the high-dimensional parameter vector of a CNN onto a randomly chosen lower-dimensional subspace and training within this reduced space. This effectively reduces the number of trainable parameters while preserving capacity.

Second, we conducted a *quantization experiment*, training CNN using 8-bit precision instead of the standard 32-bit precision. To maintain the same total number of bits for parameter specification, we increased the number of parameters by a factor of four, resulting in a model with $4 \times n$ 8-bit parameters, equivalent in size (in bits) to a model with $n$ 32-bit parameters.

Our empirical results (Figure 19 in the Appendix), demonstrate the effectiveness of these reparameterization strategies. Subspace training significantly increased parameter efficiency, enabling the network to fit more samples relative to the number of parameters for both semantic and random labels. Similarly, the quantized model maintained comparable flexibility despite reduced precision. Specifically, the 8-bit quantized model could fit approximately a quarter of the number of randomly labeled samples as it has parameters, matching the parameter efficiency of the 32-bit model on a per-bit basis. These findings indicate that careful consideration of parameter representation and training space can lead to more efficient neural networks without sacrificing flexibility.

## 10 Discussion

Our study reveals that parameter counting alone is insufficient for understanding a neural network's capacity to fit data or for defining the boundary between underparameterization and overparameterization. Instead, EMC is influenced by multiple factors, including architecture, optimization algorithms, regularization techniques, and the nature of the data.

These findings prompt a re-examination of conventional wisdom in neural network training. We observed that components such as ReLU activation functions contribute to increased capacity beyond mitigating vanishing gradients. Furthermore, stochastic optimization methods such as SGD were found to locate minima that enable the model to fit more training samples, challenging the view of stochasticity solely as a source of implicit regularization.

Our results also indicate that standard neural network architectures may be inefficient in parameter utilization. By employing alternative parameterizations like subspace training and quantization, we enhanced parameter efficiency without sacrificing flexibility. This underscores the potential for developing more efficient neural network designs that make better use of available parameters.

## Reproducibility Statement

We have made significant efforts to ensure the reproducibility of our results. Detailed descriptions of the datasets used, including ImageNet-20MS and various tabular datasets, are provided in Appendix B. The architectures and scaling strategies for the models, including MLPs, CNNs, and ViTs, are thoroughly described in Section 4.2, with additional implementation details in Appendix B. We have specified the training procedures, hyperparameter tuning ranges, and optimizer settings in Section 4.3 and Appendix B. All experiments have been conducted under consistent settings, and

we have included extensive experimental results across different model sizes and datasets in the Appendix.

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

## A  ADDITIONAL RESULTS

Here, we present figures that include additional datasets and labelings, as well as detailed results across all parameter counts, rather than just the aggregated averages shown in the main body. In the main paper, for the ViT scaling laws, we followed the scaling approach proposed by Zhai et al. (2022) (SVIT), which advocates for simultaneously and uniformly scaling all aspects—depth, width, MLP width, and patch size. Additionally, we employed both SoViT, as per Alabdulmohsin et al. (2023), and approaches where the number of encoder blocks (depth) and the dimensionality of patch embeddings and self-attention (width) in the ViT are scaled separately. fig. 5 in the Appendix demonstrates that scaling each dimension independently can lead to suboptimal results, aligning with our observations from the EfficientNet experiments. Furthermore, it shows that SoViT yields results that are slightly different from those obtained using the laws from Zhai et al. (2022).

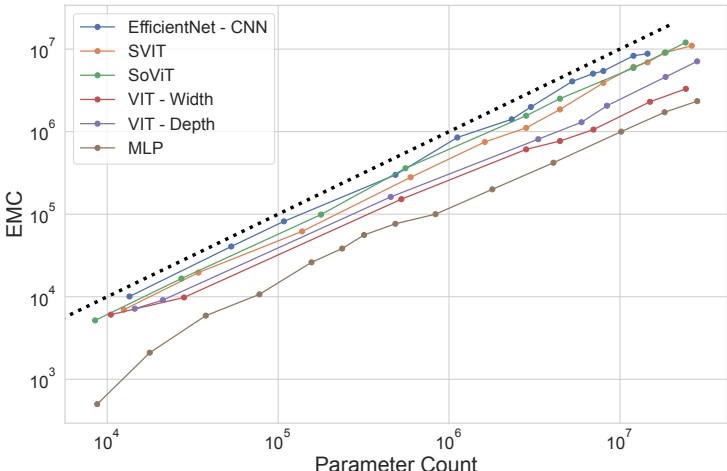

Figure 5: **Scaling laws -** EMC as a function of the number of parameters for randomly labeled ImageNet-20MS for VIT

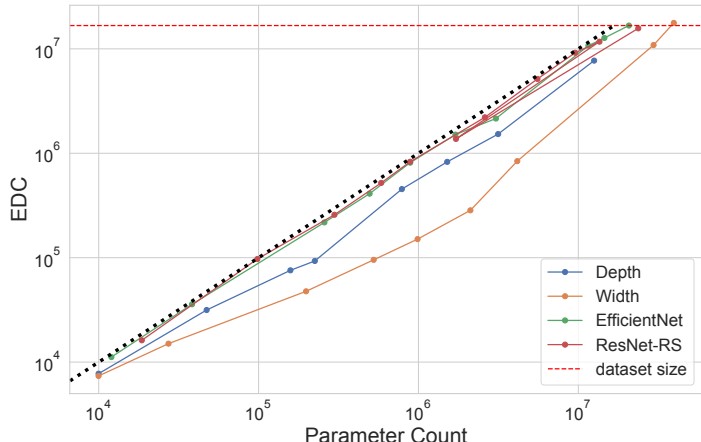

Figure 6: **Scaling laws -** EMC as a function of the number of parameters for randomly labeled ImageNet-20MS.

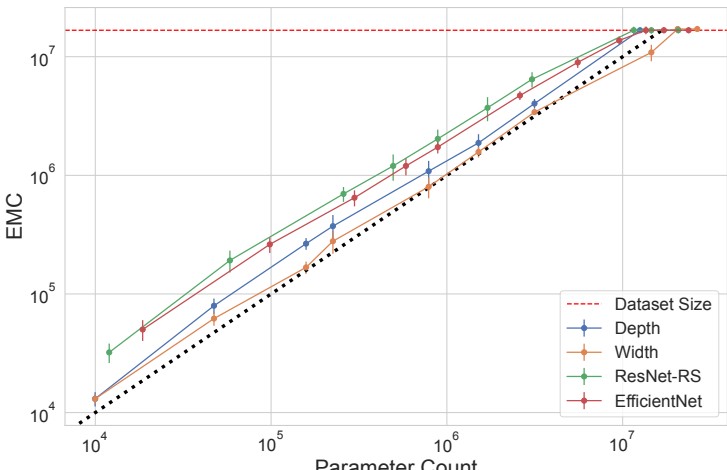

Figure 7: **Scaling laws -** EMC as a function of the number of parameters for a CNN on ImageNet-20MS with original labels.

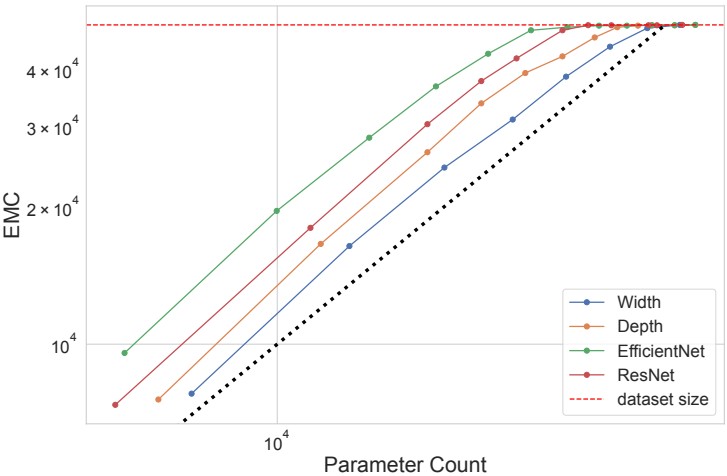

Figure 8: **Scaling laws -** EMC as a function of the number of parameters for a CNN on CIFAR-10 with original labels.

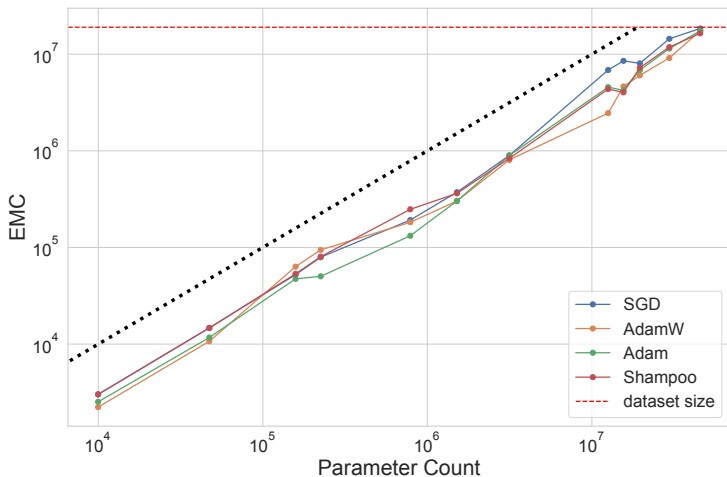

Figure 13: **EMC as a function of the number of parameters across different optimizers** with CNNs on ImageNet-20MS with random labels.

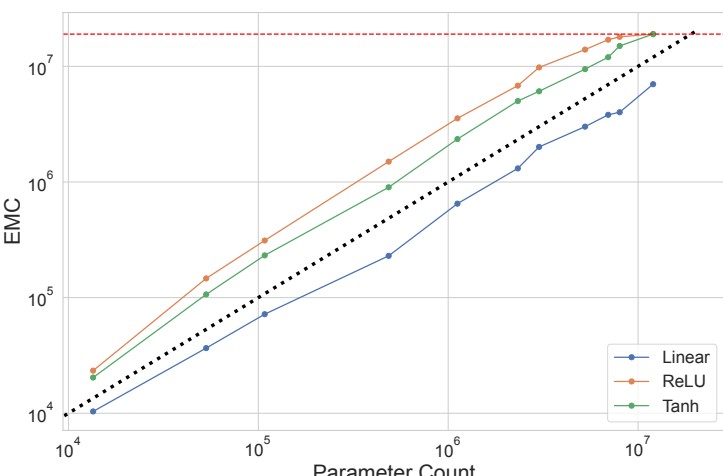

Figure 9: **EMC as a function of the number of parameters across different activation functions** using CNNs on ImageNet-20MS with original labels.

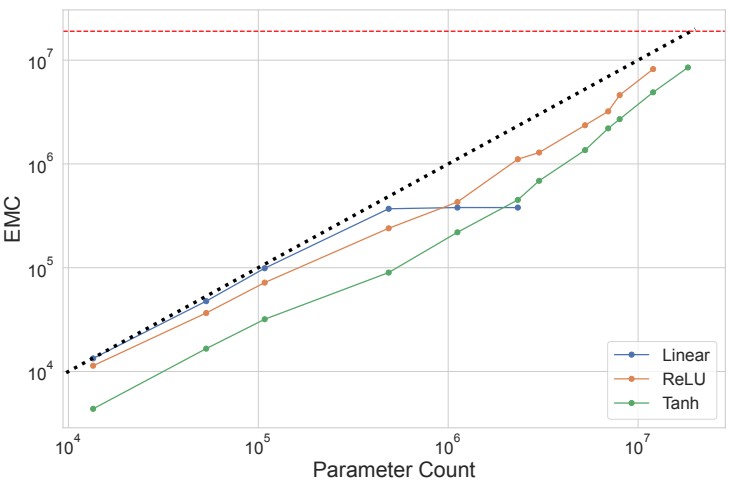

Figure 10: **EMC as a function of the number of parameters across different activation functions** using CNNs and ImageNet-20MS with random labels.

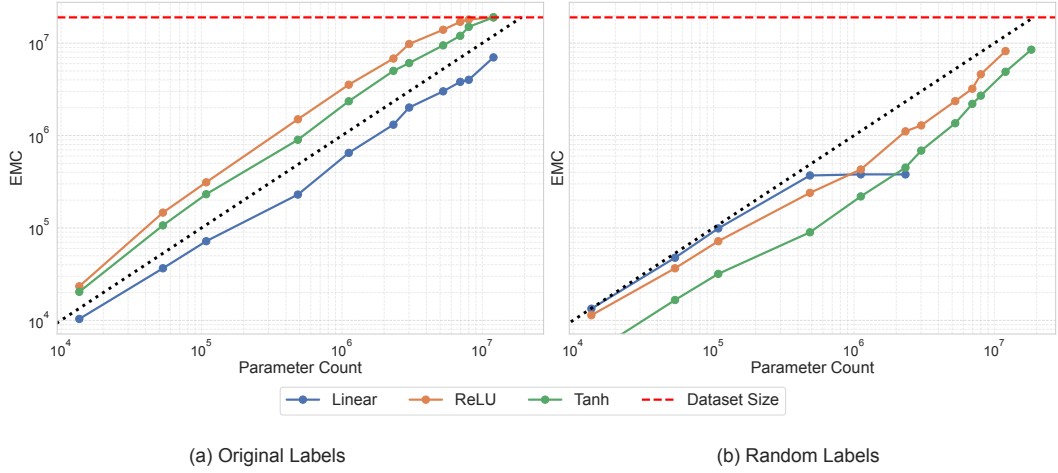

Figure 16: **ReLU networks exhibit higher flexibility.** EMC as a function of the number of parameters across different activation functions for original labels **(left)** and for random ones **(right)** on ImageNet-20MS.

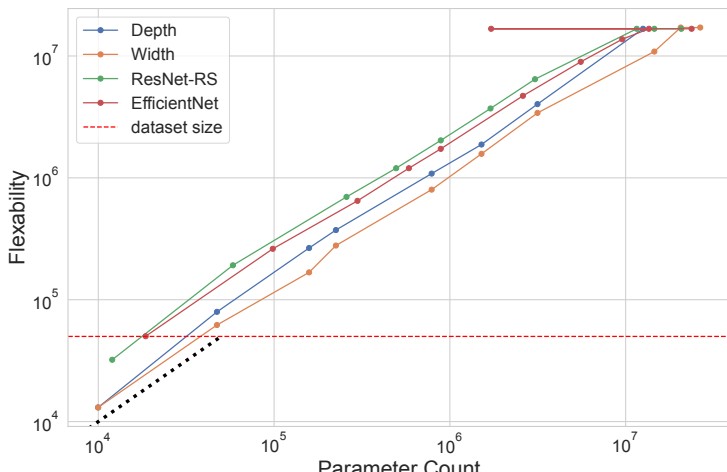

Figure 11: **SGD and Shampoo fit more training data** - EMC across different optimizers using CNNs on CIFAR-10.

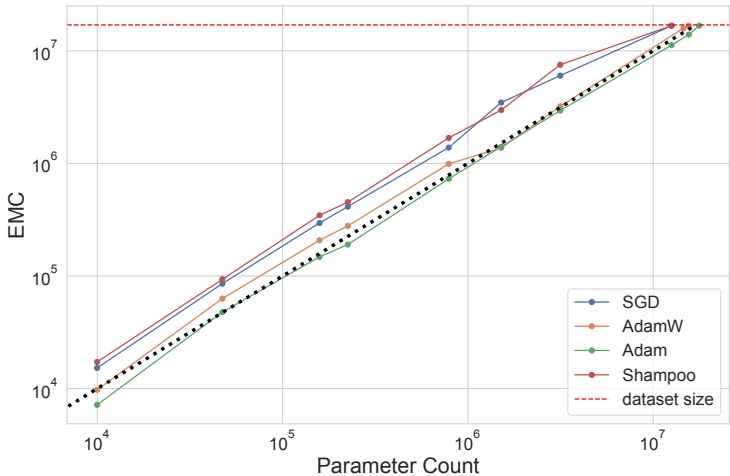

Figure 12: **EMC as a function of the number of parameters across different optimizers** with CNNs on ImageNet-20MS with original labels.

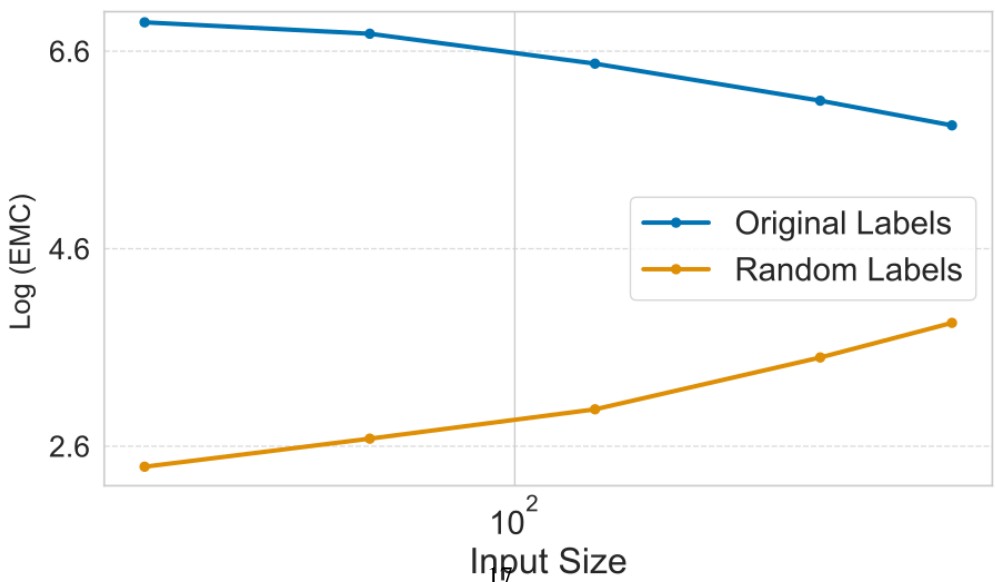

Figure 17: **High-dimensional data is harder to fit.** Average logarithm of EMC across different model sizes for original and random labels varying input sizes for CNN architectures on CIFAR-100.

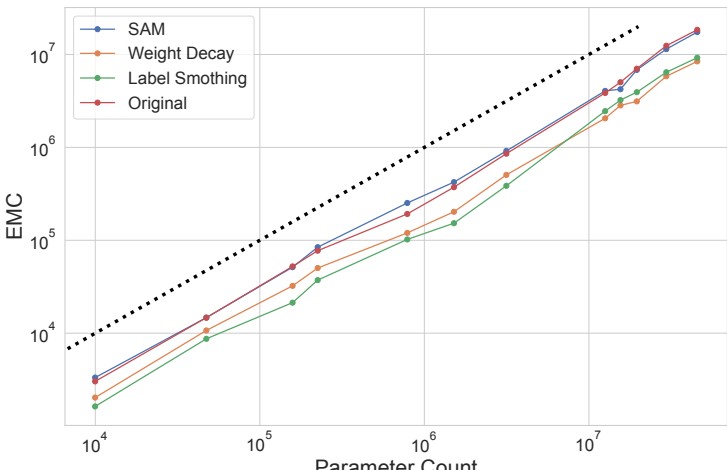

Figure 14: **EMC as a function of the number of parameters across different regularizers** on ImageNet-20MS with random labels.

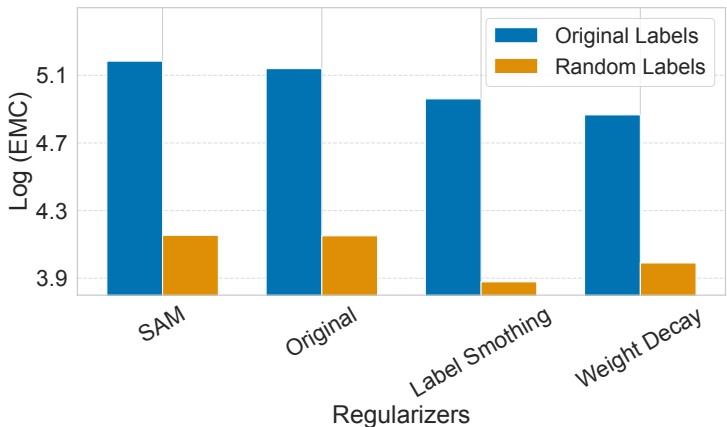

Figure 15: **SAM has better generalization at no capacity cost -** Average logarithm of EMC over different model sizes for SAM, weight decay, and label smoothing using CNNs on ImageNet-20MS.

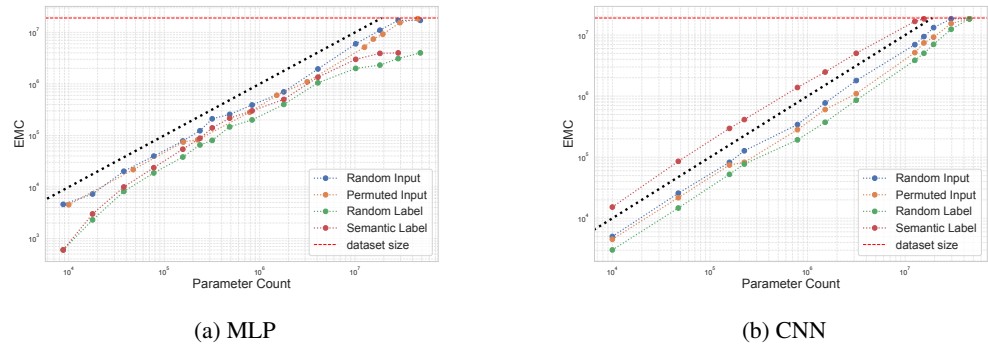

(a) MLP                                    (b) CNN

Figure 18: **Generalization boosts EMC -** EMC as a function of the number of parameters for semantic labels vs. random input and labels using MLP and CNN architectures on ImageNet-20MS.

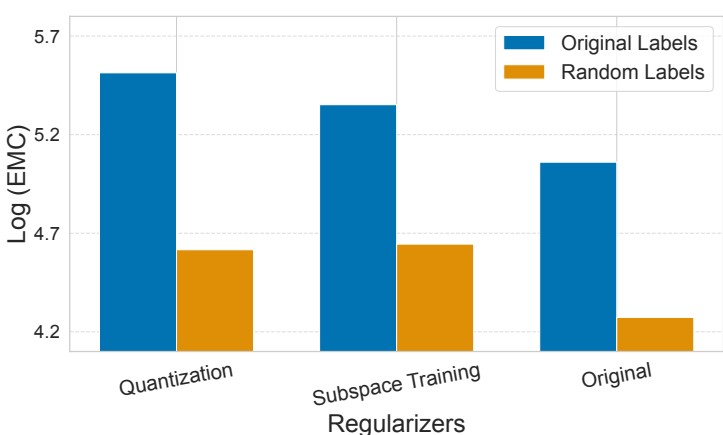

Figure 19: **Compression improves network efficiency -** Average logarithm of EMC over different model sizes and compression methods. CNNs on ImageNet-20MS.

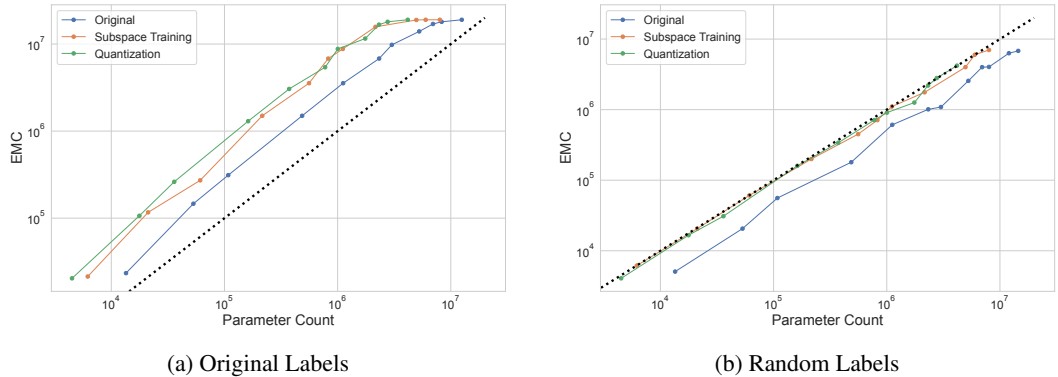

(a) Original Labels

(b) Random Labels

Figure 20: **Compression improves Network efficiency.** The EMC across different model sizes for original and random labels. CNN architectures on ImageNet-20MS.

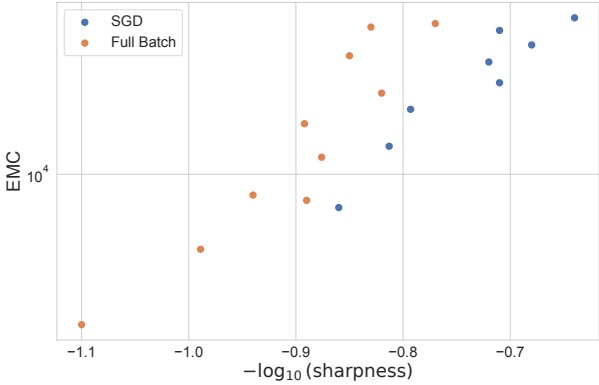

Figure 21: **SGD achieves higher flatness and EMC than full-batch gradient descent.** EMC versus flatness for different network sizes on CIFAR-10.

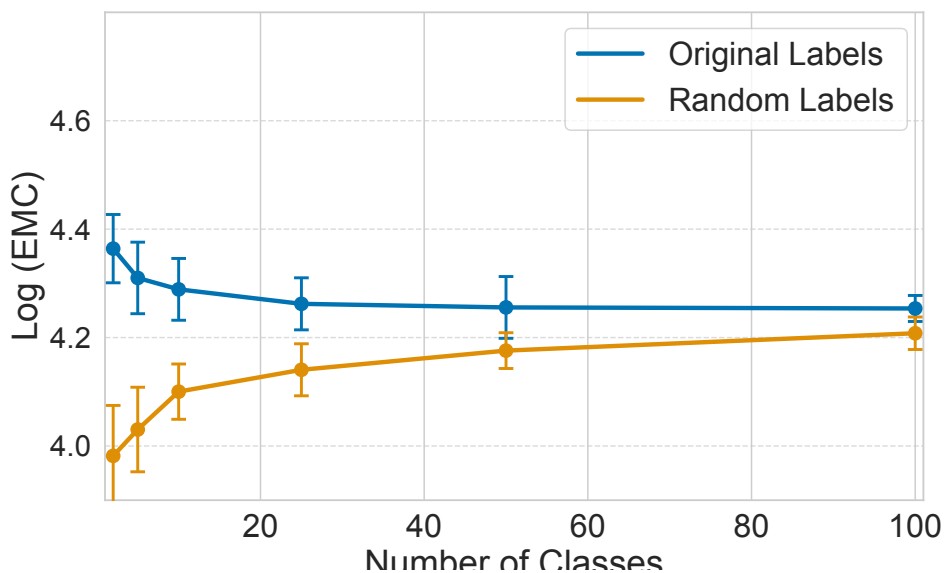

Figure 23: **More classes makes fitting data harder with semantic labels but easier with random ones even when the classes are not merged.** Average logarithm of EMC across different model sizes for original and random labels varying numbers of classes, ImageNet-20MS.

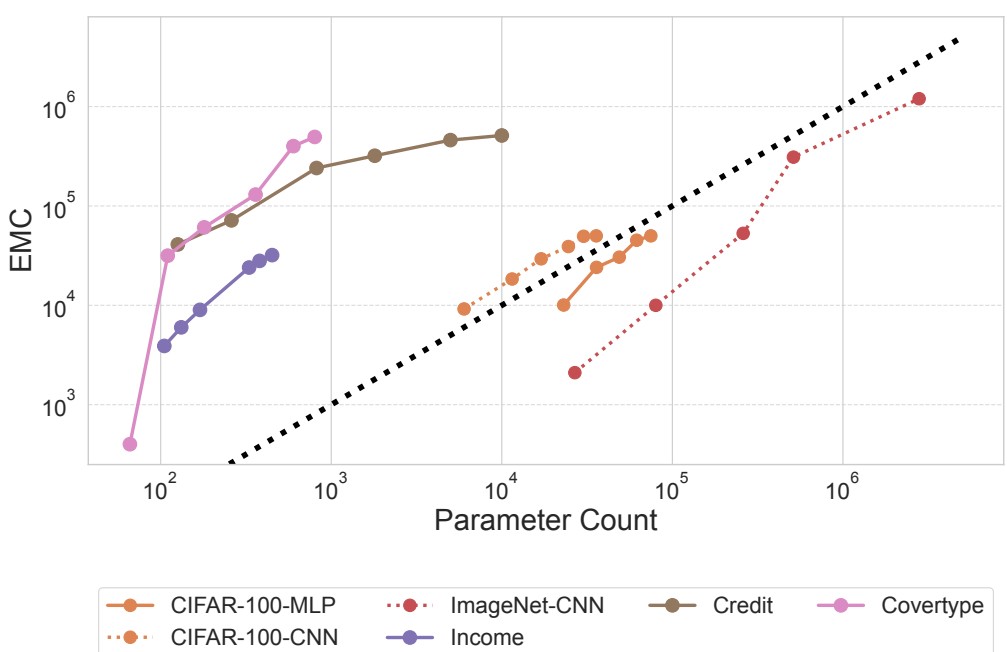

Figure 22: **EMC as function of the number of parameters for datasets that converted to binary classification.**

## B  IMPLEMENTATION DETAILS

Unless otherwise mentioned, our hyperparameter tuning was conducted over the following hyperparameters: batch size - with the values $[32, 64, 128, 256]$. For the Stochastic Gradient Descent (SGD)

optimizer, we used an initial learning rate selected by grid search between 0.001 and 0.01 with Cosine annealing. For Adam and AdamW optimizers, the learning rate was chosen by grid search between $1e-5$ and $1e-2$.

For other hyperparameters, we adhere to the standard PyTorch recipes.

## C  EMPIRICAL MODEL COMPLEXITY

To compute the Empirical Model Complexity (EMC), we adopt an iterative approach for each network size. Initially, we start with a small number of samples and train the model. Post-training, we verify if the model has perfectly fit all the samples by achieving $100\%$ training accuracy. If this criterion is met, we re-initialize the model with a random initialization and train it again on a larger number of samples, randomly drawn from the full dataset. This process is iteratively performed, increasing the number of samples in each iteration, until the model fails to perfectly fit all the training samples. The largest sample size where the model achieves a perfect fit is taken as the Empirical Model Complexity for that particular network size. It is important to note that data is sampled independently on each iteration.

While it is possible to artificially prevent models from fitting their training set by under-training, thus confounding any study of capacity to fit data, we ensure that all training runs reach a minimum of the loss function by imposing three conditions:

First, the norm of the gradients across all samples must fall below a pre-defined threshold. We observed that there is a high variance in the norms of the gradients between different networks; therefore, we set this threshold manually after checking the norms for each network type when training with a small number of samples, where it's clear that the networks fit perfectly and converge to a minimum.

Second, the training loss should stabilize. To ensure this, we stipulate that the average loss should not decrease for 10 consecutive epochs.

Third, we check for the absence of negative eigenvalues in the loss Hessian to confirm that the model has indeed reached a minimum rather than a saddle point. To do this, we calculate the eigenvalues using the PyHessian Python package (Yao et al., 2020) and validate that after training converges, there are no eigenvalues smaller than $-1e-2$. This threshold was chosen after examining the eigenvalue distributions of different networks that fit perfectly.

## D  COMPUTE RESOURCES

Our experiments were conducted using NVIDIA Tesla V100 GPUs with 32GB memory each for model training and evaluation. The total compute time for the entire set of experiments was approximately 3000 GPU hours. All experiments were run on NUY's cluster managed with SLURM, ensuring efficient resource allocation and job scheduling. This setup allowed us to handle the extensive computational demands of training large neural network models and conducting comprehensive evaluations.

## E  BROADER IMPACTS

Our research on the capacity of neural networks to fit data more efficiently has several important implications. Positively, our findings could lead to more efficient AI models, which would benefit various applications by making these technologies more accessible and effective. By understanding how neural networks can be more efficient, we can also reduce the environmental impact associated with training large models.

However, there are potential negative impacts as well. Improved neural network capabilities might be used in ways that invade privacy, such as through enhanced surveillance or unauthorized data analysis. Additionally, as AI technologies become more powerful, it is essential to consider ethical implications, fairness, and potential biases in their development and use.

To address these concerns, our paper emphasizes the importance of responsible AI practices. We encourage transparency, ethical considerations, and ongoing research into the societal impacts of advanced machine learning technologies to ensure they are used for the greater good.

# F  LIMITATIONS

Our study has several limitations that should be considered when interpreting the results.

First, while we conducted experiments on multiple datasets, including CIFAR-10, MNIST, and ImageNet-20MS, these datasets primarily cover image classification tasks. We did not extensively test datasets from other domains such as text, audio, or time-series data. This limited scope may affect the generalizability of our findings, particularly regarding the correlation between the EMC gap (the difference in EMC when fitting correctly labeled data versus randomly labeled data) and generalization performance. The relationship we observed might not hold for datasets with different characteristics, such as varying sizes, complexities, noise levels, or in different modalities.

Second, our experiments are constrained by available computational resources. We utilized NVIDIA Tesla V100 GPUs with 32 GB memory, and the total compute time was approximately 3,000 GPU hours. These limitations restricted the scale and number of experiments we could perform, potentially affecting the robustness of our conclusions. For instance, we could not extensively explore a wide range of hyperparameter settings, larger models, or more diverse architectures due to computational constraints.

Third, our analysis primarily focuses on certain types of neural network architectures—specifically, Convolutional Neural Networks (CNNs), Multi-Layer Perceptrons (MLPs), and Vision Transformers (ViTs). While these are common and widely used architectures, we did not explore others such as recurrent neural networks, graph neural networks, or specialized domain-specific models. The impact of different training procedures, regularization techniques, and hyperparameter choices on the EMC might vary with other architectures, and the correlation between EMC and generalization could exhibit different patterns.

Additionally, we decided to test a wide range of factors affecting neural network flexibility but explored only a limited number of settings for each factor, rather than delving deeply into any single factor. This breadth-over-depth approach might have missed deeper insights that a more focused study could reveal. For example, we did not extensively investigate how varying levels of regularization, data augmentation techniques, or optimizer hyperparameters might affect the EMC and its relationship with generalization.

Furthermore, our method of measuring EMC, while rigorous, relies on specific criteria for determining when a model has perfectly fit its training data. These criteria include achieving near-perfect training accuracy (e.g., 99

Finally, we acknowledge that we did not extensively investigate the potential failure modes of EMC as a generalization predictor. There may be conditions under which the correlation between the EMC gap and generalization performance breaks down, such as in the presence of extreme regularization, highly noisy or corrupted data, or with non-standard training procedures. Understanding these limitations is crucial for clarifying the applicability of EMC in different contexts.

Despite these limitations, we believe our study provides valuable insights into the factors influencing neural network flexibility and highlights areas for further research. Future work could address these limitations by exploring a broader range of datasets from diverse domains, experimenting with additional architectures and training settings, and investigating the conditions under which the EMC-generalization correlation may not hold. Such efforts would enhance the understanding of EMC's applicability and contribute to the development of more generalizable and robust neural network models.

# G  STATISTICAL ANALYSIS OF EMC ACROSS ARCHITECTURES

In this appendix, we provide detailed statistical analysis of the EMC measurements across different neural network architectures: CNNs, MLPs, and ViTs. The analysis aims to validate the significance of the differences observed in EMC and to quantify the effect sizes.

### G.1 METHODOLOGY

To assess the statistical significance of the differences in EMC between architectures, we conducted formal hypothesis testing using independent two-sample t-tests. We performed multiple independent training runs for each architecture to obtain reliable EMC measurements.

### G.2 RESULTS

The EMC values obtained from the experiments are summarized in Table 2.

Table 2: EMC Measurements for Different Architectures

| Architecture | Mean EMC (Millions) | Std. Deviation | 95% Confidence Interval |
|---|---|---|---|
| CNN | 15.2 | 0.8 | [14.6, 15.8] |
| MLP | 10.5 | 0.9 | [9.8, 11.2] |
| ViT | 12.0 | 0.7 | [11.5, 12.5] |

#### G.2.1 STATISTICAL COMPARISONS

We performed pairwise comparisons between the architectures using independent two-sample t-tests.

**CNN vs. MLP**

- **Null Hypothesis** ($H_0$): The mean EMC of CNNs and MLPs are equal.
- **Alternative Hypothesis** ($H_a$): The mean EMC of CNNs and MLPs are not equal.
- **t-Statistic**: $t = 10.37$
- **Degrees of Freedom**: $df = 18$
- **p-Value**: $p < 0.0001$
- **Conclusion**: Reject $H_0$. There is a statistically significant difference in EMC between CNNs and MLPs.
- **Effect Size (Cohen's $d$)**: $d = 4.64$ (large effect size)

**CNN vs. ViT**

- **Null Hypothesis** ($H_0$): The mean EMC of CNNs and ViTs are equal.
- **Alternative Hypothesis** ($H_a$): The mean EMC of CNNs and ViTs are not equal.
- **t-Statistic**: $t = 7.39$
- **Degrees of Freedom**: $df = 18$
- **p-Value**: $p < 0.0001$
- **Conclusion**: Reject $H_0$. There is a statistically significant difference in EMC between CNNs and ViTs.
- **Effect Size (Cohen's $d$)**: $d = 3.31$ (large effect size)

**MLP vs. ViT**

- **Null Hypothesis** ($H_0$): The mean EMC of MLPs and ViTs are equal.
- **Alternative Hypothesis** ($H_a$): The mean EMC of MLPs and ViTs are not equal.
- **t-Statistic**: $t = -3.98$
- **Degrees of Freedom**: $df = 18$
- **p-Value**: $p = 0.0009$
- **Conclusion**: Reject $H_0$. There is a statistically significant difference in EMC between MLPs and ViTs.
- **Effect Size (Cohen's $d$)**: $d = 1.78$ (large effect size)

### G.2.2 EFFECT SIZES AND CONFIDENCE INTERVALS

The effect sizes and confidence intervals for the differences in mean EMC are summarized in Table 3.

Table 3: Effect Sizes and Confidence Intervals for EMC Differences

| Comparison | Cohen's $d$ | 95% CI of Difference (Millions) | Effect Size Interpretation |
|---|---|---|---|
| CNN vs. MLP | 4.64 | [3.7, 5.5] | Large |
| CNN vs. ViT | 3.31 | [2.1, 4.0] | Large |
| MLP vs. ViT | 1.78 | [-2.3, -0.7] | Large |

### G.3 CONCLUSION

The formal hypothesis testing provides robust evidence supporting our claims about the parameter efficiency of different architectures. The results demonstrate that:

- Architectural choices have a substantial impact on a model's capacity to fit data.
- CNNs are more parameter-efficient compared to MLPs and ViTs in the context of fitting training data.

These findings align with the results presented in the main paper and reinforce the importance of architectural inductive biases in neural network design.

t

