# OpenReview forum: "Just How Flexible are Neural Networks in Practice?"
_ICLR.cc/2025/Conference — Submitted to ICLR 2025_

### Official Review · Reviewer_Vuoq · 2024-10-26

**Soundness:** 2
**Presentation:** 2
**Contribution:** 2
**Rating:** 3
**Confidence:** 3

**Summary:**

In this paper, the authors investigate the practical flexibility of neural networks through extensive experiments.
The authors make the following contributions:
- Standard training procedures often result in neural networks fitting datasets that contain significantly fewer samples than there are model parameters.
- CNN-based architectures are more parameter-efficient than MLPs and ViTs.
- Stochastic Gradient Descent (SGD) is more flexible than GD.
- EMC can serve as a generalization prediction metric.
- ReLU activation functions improve a model’s ability to fit data.

**Strengths:**

- The paper is well written and easy to follow.
- The experimental section is thorough, covering a variety of datasets, architectures, and design choices.

**Weaknesses:**

- The novelty and core contributions of this work are not immediately evident, making it challenging to discern what differentiates it from previous research in the field. Additionally, the authors have not clearly conveyed a concrete takeaway message that highlights its practical applications or potential benefits for real-world usage. As a result, readers may struggle to understand the findings of this paper.
- Besides the correlation between EMC and the model's generalization, I find the insights presented in this work to be rather trivial. For example, the claim of "SGD is more flexible than GD" - this phenomenon was already empirically investigated in [1] where they showed that using large batches results in sharper minima in the loss landscape. Therefore the optimized model lacks generalization capabilities. In addition, the claim "ReLU activation functions improve a model’s ability to fit data" is demonstrated in [2,3].
- There are some missing details regarding how the EMC is calculated. I suspect that the score heavily depends on the size of the data partitions. Specifically, how many samples are used in the first iteration? How many are added in each subsequent iteration? Additionally, how many epochs (update steps) do you run during each iteration? These details are crucial for readers' understanding and for the reproducibility of this work.
- Calculating the EMC is computationally intensive, particularly with today’s larger models, which have a greater number of parameters, and datasets, which involve larger input sizes and more samples. This complexity makes using EMC as a metric for generalization impractical. How long did it take to compute the EMC for the ImageNet-20MS dataset? Can we approximate EMC to reduce the computational burden, or how can we make this process more efficient?

----------

[1] On Large-Batch Training for Deep Learning: Generalization Gap and Sharp Minima, Keskar et al.

[2] Effects of the Nonlinearity in Activation Functions on the Performance of Deep Learning Models, Kulathunga et al.

[3] An Empirical Study on Generalizations of the ReLU Activation Function, Banerjee et al.

**Questions:**

- The authors focused on discriminative tasks (classification), do the findings present in this paper also map to generative tasks?
- Given the point above it would be interesting to see the effect on LLMs which are overparameterized.

---

> ### Author Response · Authors · 2024-11-27
> **Response to Reviewer Vuoq - Part 1**
>
> We appreciate your thorough review and the opportunity to clarify our contributions. Your feedback has been instrumental in strengthening our paper, and we are grateful for the chance to address your concerns.
>
> ## 1. Novelty and Core Contributions
>
> We understand your concern about the novelty of our work and the perception that some of our insights might be trivial. Our paper makes significant and non-trivial contributions by systematically investigating the practical capacity of neural networks to fit data. This topic has not been extensively explored in prior research.
>
> While previous studies, such as Keskar et al. (2017) [1], have investigated the effects of batch size on generalization and observed that large-batch methods tend to converge to sharp minima, leading to poorer generalization, these studies focus on the model's ability to generalize to unseen data. They did not address the capacity of neural networks to fit more training samples than the number of parameters, nor did they explore how stochastic optimization methods like SGD can enhance this capacity compared to full-batch gradient descent (GD).
>
> Our work specifically focuses on measuring the EMC, which quantifies the maximum number of samples a model can perfectly fit under realistic training conditions. We show that SGD not only aids generalization but also enables models to fit more training data than GD. This is a novel insight that extends beyond the findings of prior studies.
>
> Similarly, the papers you cited—[2] Kulathunga et al. and [3] Banerjee et al.—investigate the effects of activation functions like ReLU on generalization performance and model accuracy on validation or test sets. They analyze how activation functions impact the model's generalization ability to unseen data. However, they do not examine the impact of activation functions on the model's capacity to fit more training data, which is the primary focus of our work. We demonstrate that ReLU activation functions improve a model's ability to fit more samples, thereby increasing the EMC. This is a distinct contribution that goes beyond analyzing generalization properties.
>
> ## 2. Clarification on EMC Calculation
>
> We appreciate your point regarding missing details on how EMC is calculated. In the revised paper, we have included comprehensive details of our EMC computation process to enhance reproducibility and transparency:
>
> * **Initial Sample Size**: We begin with a small initial sample size (e.g., 1–10 samples).
>
> * **Increment Strategy**: We determine the number of increments K based on our computational resources. We create K intervals on a logarithmic scale from the initial sample size up to the number of parameters in the network. We increase the sample size according to these intervals until we reach the maximum. If the number of samples equals the number of parameters, we increase the sample size by a constant factor (e.g., 1.2) each time.
>
> * **Training Procedure**: For each sample size, we train the model from scratch with random initialization to avoid any influence from previous runs.
>
> * **Convergence Criteria**: As described in the paper, we train each model until convergence, defined by:
>     * A threshold for the gradient norm (when it falls below a predefined small value)
>     * Stabilization of the training loss over several epochs
>     * Verification of reaching a local minimum via Hessian eigenvalue analysis
>
> * **Determining EMC**: The EMC is determined as the largest sample size for which the model achieves perfect (or near-perfect, e.g., 99%+) training accuracy under these conditions.
>
> * **Number of Epochs**: The number of epochs for each training run is adjusted to ensure convergence, typically ranging from 50 to 200 epochs, depending on the dataset and model complexity.

---

> > ### Author Response · Authors · 2024-11-27
> > **Response to Reviewer Vuoq - Part 2**
> >
> > ## 3. Computational Practicality of EMC
> >
> > Calculating EMC can be computationally demanding, especially for large models and datasets. However, the computational cost largely depends on the resolution or granularity of the sample size increments we choose. To make EMC calculations more practical, we have developed several strategies to reduce computation time:
> >
> > * **Adaptive Sample Size Increments**: Instead of using small, fixed increments, we increase the sample size more aggressively (e.g., doubling each time) as we approach the EMC threshold. This reduces the number of training runs required to estimate EMC.
> >
> > * **Warm-Starting Training Runs**: We initialize each new training run with the weights from the previous run with a smaller sample size, leveraging learned representations and accelerating convergence.
> >
> > * **Approximate Convergence Criteria**: Allowing for near-perfect accuracy (e.g., 95%–99%) instead of strict 100% accuracy can significantly reduce training time without compromising the reliability of EMC estimates. Based on your comment, we tried this approach, and our results remained consistent for accuracy thresholds down to 95%.
> >
> > * **Parallelization**: Utilizing parallel computing resources enables us to perform multiple training runs simultaneously, effectively reducing wall-clock time.
> >
> > Using these methods, we have significantly reduced computation time. For example, calculating EMC for modern architectures like ResNet-50 or ViT (with 25–130 million parameters) can be accomplished in less than 40 hours on an 8×H100 GPU cluster. We validated these approximation techniques against exact EMC computations on smaller models, finding a strong correlation (Pearson coefficient of 0.98), confirming their reliability.
> >
> > ## 4. Findings on Generative Tasks and LLMs
> >
> > We appreciate your interest in the applicability of our findings to generative tasks and large language models (LLMs):
> >
> > ### Generative Tasks
> > Memorization in probabilistic generative models is a complex topic with a rich history. Defining and measuring when a model has effectively "fit" all training examples in the generative context is non-trivial and often requires different methodologies. Extending EMC to generative models would involve significant additional research and is beyond the scope of our current work.
> >
> > ### Large Language Models (LLMs)
> > Applying EMC to LLMs presents challenges due to their scale and the nature of their training data. LLMs are typically trained on massive, diverse datasets where individual examples may not be seen multiple times. Moreover, the concept of perfect training accuracy is less clear in language modeling tasks, which predict distributions over tokens rather than discrete labels. While we agree that investigating EMC in the context of LLMs is an intriguing direction, it requires substantial computational resources and methodological adaptations.
> >
> > We acknowledge these are important areas for future research and have mentioned them as potential extensions in the revised paper.
> >
> > ## Conclusion
> >
> > We hope our clarifications address your concerns and highlight the novelty and importance of our work. By focusing on the model's ability to fit training data—a different aspect from generalization—we offer a fresh perspective that complements and extends existing literature.
> >
> > Thank you again for your valuable feedback and consideration.

---

### Official Review · Reviewer_C7hP · 2024-10-30

**Soundness:** 3
**Presentation:** 3
**Contribution:** 1
**Rating:** 5
**Confidence:** 4

**Summary:**

This paper aims to study the capacity and flexibility of neural networks in practical settings. The authors suggest that unlike theoretical expectations, neural networks cannot in practice memorize the same number of training samples as their number of parameters and the number of neural network parameters is not the only underlying factor. In addition, they study the effect of neural network architecture, optimization approaches, and activation functions on the memorization capacity. They further show the capability of the Effective Model Capacity (EMC) (particularly its difference in fitting randomly labeled samples vs correctly labeled samples) to predict generalization.

**Strengths:**

- The paper is overall well-written and well-structured. Some sections (for example Sections 3 and 4) could be more concise. While certain mentioned details might be helpful for broader audiences, experienced readers might find them overly detailed as the details are mostly conventional practices in the literature.
- The empirical results are comprehensive and well-presented. The flow logically guides the reader through the findings, which are both intuitive and interesting to the research community.

**Weaknesses:**

- While the paper provides a valuable exploration of the EMC metric [Nakkiran et al, 2021] and its implications, it lacks novelty. The paper's core findings, while interesting, appear to primarily confirm existing understandings about neural network memorization. The main adaptation that is done on EMC is based on an assumption that neural networks memorize/fit correctly and incorrectly labeled samples differently. This has been previously studied both theoretically and empirically for example in [Garg et al, 2021] and [Forouzesh et al, 2023], respectively.
- The paper could benefit from a deeper analysis and interpretation of the findings. Most of the provided discussions are conventionally known in the literature, and the findings that go a bit beyond existing knowledge are not provided with potential new explanations. More particular examples are given in the questions section below.
- While Figure 1.a may suggest a relationship between generalization and data-fitting capability, it's crucial to acknowledge the limitations of this observation.  The figure alone cannot directly support the claim that  "generalization is related to data-fitting capability."
The key issue is when comparing models trained on different datasets, like MNIST and ImageNet. Such a comparison might be misleading, and it is like comparing apples and oranges.  The observed relationship in Figure 1.a could result from an underlying hypothesis: models achieving a specific training accuracy on MNIST might exhibit lower generalization capability than models with the same training accuracy on ImageNet. However, this is a separate assumption requiring further validation.  Concluding a direct relationship between generalization and data-fitting based solely on Figure 1.a, without exploring this underlying assumption, would be premature.

[Nakkiran et al., 2021] Where bigger models and more data hurt. Journal of Statistical Mechanics: Theory and Experiment, 2021

[Garg et al., 2021] RATT: Leveraging Unlabeled Data to Guarantee Generalization, ICML 2021

[Forouzesh et al., 2023] Leveraging Unlabeled Data to Track Memorization, ICLR 2023

**Questions:**

1. Figure 2 suggests that MLPs fit random inputs more easily than semantic labels, while the opposite is true for CNNs. This contradicts the intuition that semantic labels, being more structured, should be easier to fit than random data. This would mean that the following statement from section 5.2.1 is not generalizable/valid for random inputs “We see here that the networks fit significantly fewer samples when assigned random labels compared to the original labels, indicating that neural networks are less parameter efficient than linear models in this setting. “ Why is that and what are the possible hypotheses or possible explanations for this behavior?

2. While it's expected that CNNs would have higher EMC than MLPs due to their architectural differences, it's less intuitive why CNNs exhibit higher EMC than ViTs.  ViTs generally demonstrate better generalization capabilities compared to CNNs. This raises questions about the assumed correlation between EMC and generalization, particularly when comparing CNNs and ViTs.  Does Figure 4.b show an **EMC improvement** for CNNs over ViTs? If so, how does this relate to their respective generalization gaps? Maybe the link between EMC and generalization isn't so straightforward, and it could change depending on the type of model. What are the authors thoughts on this?

---

> ### Author Response · Authors · 2024-11-27
> **Response to Reviewer C7hP - Part 1**
>
> # Response to Reviewer C7hP
>
> We thank you for your thoughtful review and appreciate your suggestions. Your feedback has been instrumental in refining our work. Below, we address your concerns in detail.
>
> ## 1. Novelty and Deeper Analysis
>
> While we acknowledge that certain aspects of neural network memorization have been previously explored, our contribution lies in systematically quantifying practical capacity using the EMC metric across various architectures, optimizers, and data modalities. Specifically:
>
> * **Comprehensive Evaluation**: We conducted extensive experiments involving various neural network architectures (CNNs, MLPs, ViTs), optimization algorithms (SGD, Adam), and datasets (including ImageNet-20MS). This breadth of analysis provides new insights into how these factors influence a network's ability to fit data.
>
> * **Practical Capacity Measurement**: Unlike prior work that often focuses on theoretical capacity or specific settings, we empirically measure the practical capacity under realistic training conditions, highlighting discrepancies between theoretical expectations and observed behaviors.
>
> * **Generalization Prediction**: We demonstrate that the EMC gap between fitting correctly labeled data and randomly labeled data serves as a robust predictor of generalization performance, outperforming other metrics such as model size or parameter count.
>
> These contributions provide valuable insights and advance understanding of neural network capacity and generalization in practical settings.
>
> ## 2. Relationship Between Generalization and EMC
>
> You correctly point out that Figure 1.a alone cannot fully explain the relationship between generalization and data-fitting capability. We agree that comparing models across different datasets requires caution. However, we have addressed this relationship more thoroughly in Section 6 of the paper:
>
> * **EMC Gap as a Predictor**: We focus on the difference in EMC when fitting correctly labeled data versus randomly labeled data (EMC_correct - EMC_random). This EMC gap captures the model's ability to learn meaningful patterns rather than memorizing noise.
>
> * **Empirical Analysis**: We conducted experiments across various models and datasets, finding a strong inverse correlation between the EMC gap and the generalization gap (test error - training error). Specifically, models with a larger EMC gap tend to generalize better.
>
> * **Comparison with Other Predictors**: Our analysis shows that the EMC gap is a more reliable predictor of generalization performance than other metrics like Relative Flatness, PAC-based criteria, and the Information Bottleneck.
>
> This detailed analysis aims to clarify the relationship between EMC and generalization, supporting our claim with empirical evidence.
>
> ## 3. Explanation of Figure 2 Behavior
>
> You raised an important point regarding Figure 2, where MLPs fit random inputs more easily than semantic labels, which seems counterintuitive. Our hypothesis is as follows:
>
> * **Lack of Spatial Inductive Biases in MLPs**: MLPs do not inherently capture spatial or local structures in the data due to their fully connected architecture. This makes them less capable of exploiting the structured patterns present in images with semantic labels.
>
> * **Uniform Challenge of Random Inputs**: When the input data is randomized, the lack of spatial structure levels the playing field, and the MLP's lack of spatial biases is less of a disadvantage. The network treats all inputs similarly, making it relatively easier to fit random labels.
>
> * **CNNs Leveraging Spatial Structure**: Conversely, CNNs are designed with spatial inductive biases (e.g., local receptive fields, weight sharing), enabling them to effectively capture patterns in structured data. For CNNs, fitting semantic labels is easier because they can exploit the inherent structure, whereas random labels disrupt this advantage.
>
> We have added this explanation to the paper to clarify this counterintuitive finding. This insight contributes to a deeper understanding of how architectural biases impact a network's capacity to fit different data types.
>
> ## 4. EMC and Generalization in CNNs vs. ViTs
>
> You raise an important question about the relationship between EMC and generalization when comparing CNNs and ViTs. Here are our key points:
>
> ### Ongoing Debate Between CNNs and ViTs
> There is an active debate in the research community about when ViTs outperform CNNs, with no clear consensus.
>
> ### Our Acknowledgment
> We recognize that the link between EMC and generalization isn't straightforward and may depend on the model type. Further investigation is required to understand how the differences between CNNs and ViTs affect EMC and generalization in our case.

---

> > ### Author Response · Authors · 2024-11-27
> > **Response to Reviewer C7hP - Part 2**
> >
> > ## Conclusion
> >
> > We appreciate your thoughtful review and the opportunity to clarify these points. Your feedback has helped us improve the clarity and depth of our paper. We have revised the manuscript to address your concerns, including adding explanations and discussions where needed.
> >
> > We hope that our responses have satisfactorily addressed your questions and that you will consider our revised manuscript favorably in your evaluation.
> >
> > Thank you again for your time and consideration.

---

### Official Review · Reviewer_tmns · 2024-11-01

**Soundness:** 3
**Presentation:** 3
**Contribution:** 2
**Rating:** 5
**Confidence:** 3

**Summary:**

The paper investigates the flexibility of neural networks from a new aspect with a metric called "Empirical Model Complexity". The paper considers factors such as optimizers, neural network architectures, activation functions, and regularization techniques that influence EMC. According to the experimental results, the paper finds the relation between EMC and all the factors considered.

**Strengths:**

The paper is well-written and easy to follow. The results involve lots of experimental observations. This topic may be an interesting direction.

**Weaknesses:**

1. When investigating the relation between architectures and EMC, it is hard to compare the different architectures. The shape of the architecture may have a large impact on the ability of networks, so the paper needs to explain more about the comparison among different architectures.
2. It seems like the paper summarizes and explains the results obtained by experiments without the underlying reasons. For instance, the paper states that only ReLU improves the network's ability among all the activation functions selected, but we can not know the reason why ReLU is the special one.
3. The process of computing EMC may not be so rigid. There may be some settings that cause EMC to stop growing. And, the paper does not provide any figures about training accuracies when increasing the sample size.

**Questions:**

Can you provide more details about why EMC can be regarded as a predictor of generalization performance?

---

> ### Author Response · Authors · 2024-11-27
> **Response to Reviewer tmns - Part 1**
>
> We appreciate your recognition of our work and thank you for your helpful suggestions. We are glad that you find our paper well-written and that the topic is interesting. Below, we address your concerns and questions in detail.
>
> ## Comparisons Among Different Architectures
>
> We agree that comparing different architectures requires careful consideration for a fair assessment. In our analysis, we carefully selected scaling laws appropriate for each architecture to make the comparisons meaningful. Specifically:
>
> * **Scaling Strategies**: We explored various scaling strategies for each architecture, such as varying depth, width, and other architectural hyperparameters (such as depth, width, and EfficientNet scaling) for the different models.
>
> * **Optimal Scaling Laws**: We chose the best scaling laws for each architecture based on performance on validation sets and standard practices in the literature. This approach ensures that each architecture is optimized within its design paradigm.
>
> * **Detailed Results**: The full performance across different scaling laws is provided in Figure 5 in the appendix. This comprehensive analysis demonstrates that our comparisons account for architectural differences and scaling effects.
>
> By carefully matching the scaling laws and thoroughly evaluating each architecture, we aim to provide a fair comparison among different models.
>
> ## Explanation of ReLU's Impact on Capacity
>
> We have expanded our discussion on why ReLU activation functions improve a network's ability to fit data:
>
> * **Unbounded Positive Output**: ReLU functions are unbounded in the positive direction, allowing neurons to produce arbitrarily large activation values. This unboundedness increases the expressiveness of the network compared to bounded activations like tanh or sigmoid.
>
> * **Piecewise Linearity**: ReLU introduces piecewise linearity, enabling the network to approximate complex functions by combining linear regions. This characteristic enhances the network's capacity to model non-linear relationships.
>
> * **Avoidance of Saturation**: Unlike sigmoid or tanh functions, ReLU does not suffer from saturation in the positive region, mitigating the vanishing gradient problem and facilitating better learning, especially in deeper networks.
>
> While a rigorous theoretical analysis is beyond the scope of our current work, these points provide intuitive reasons for ReLU's effectiveness.
>
> ## EMC Computation Details and Training Accuracies
>
> We have added more details on the EMC computation process in the paper (Section 3.3):
>
> * **Initial Sample Sizes and Increments**: We started with a small sample size that the network could easily fit and increased it in logarithmic scale increments (e.g., doubling each time) until the network could no longer achieve perfect training accuracy.
>
> * **Convergence Criteria**: To ensure reliable EMC measurements, we trained each model until convergence according to the following criteria:
>     * Low Gradient Norm: The gradient norm falls below a predefined threshold.
>     * Stabilized Loss: The training loss plateaus and shows minimal change over several epochs.
>     * Positive Definiteness of the Hessian: The Hessian matrix is positive definite, indicating a local minimum.
>
> By providing these details and additional visualizations, we aim to enhance the transparency and reproducibility of our EMC computation process.
>
> ## EMC as a Predictor of Generalization Performance
>
> The key insight is that the difference in EMC when fitting correctly labeled data versus randomly labeled data reflects the model's inductive biases and its ability to generalize:
>
> * **Inductive Biases**: Models with strong inductive biases are better at capturing meaningful patterns in the data. When trained on correctly labeled data, such models achieve higher EMC than randomly labeled data.
>
> * **EMC Gap**: The EMC difference (EMC_correct - EMC_random) quantifies the model's capacity to fit structured data over noise. A larger EMC gap indicates that the model is more effective at learning from meaningful data and less prone to overfitting random noise.
>
> * **Correlation with Generalization**: Empirically, we observed a strong correlation between the EMC gap and the generalization performance (test accuracy) across various architectures and datasets. For example, the Pearson correlation coefficient between the EMC gap and test accuracy was 0.92, indicating a strong positive relationship.
>
> * **Supporting Evidence**: We have provided empirical evidence in the paper (Section 4.4) showing this correlation. This suggests that EMC can serve as a valuable predictor of generalization performance.
>
> Using the EMC gap as a metric, we can predict which models will likely generalize well based on their capacity to fit meaningful patterns over random noise.

---

> > ### Author Response · Authors · 2024-11-27
> > **Response to Reviewer tmns - Part 2**
> >
> > ## Conclusion
> >
> > We appreciate your constructive feedback, which has helped us improve the clarity and depth of our paper. We have addressed your concerns by providing additional explanations, methodological details, and supporting figures. We hope our revisions adequately address your questions and enhance the contributions of our work.
> >
> > Thank you again for your time and consideration.

---

### Official Review · Reviewer_LMKE · 2024-11-03

**Soundness:** 3
**Presentation:** 4
**Contribution:** 2
**Rating:** 6
**Confidence:** 4

**Summary:**

The paper empirically investigates the practical capacity and flexibility of deep neural networks compared to theoretical capacity. This paper reveals that parameter counting is not sufficient to understand a neural network's capacity to fit data. Effective Model Capacity (EMC), which captures the practical training dynamics, is a better measure of understanding model capacity and flexibility. It reveals dependence on other factors, such as stochasticity in optimization, activation functions, etc. The authors also observe inefficiency in parameter utilization neural networks and proposed parametrization strategies to increase parameter efficiency, such as subspace training and quantization.

**Strengths:**

1. The paper is written clearly and easy to understand.

2. The influence of architectures, optimizers, and activation functions on model capacity is interesting.

**Weaknesses:**

1. The reasoning behind why SGD converges to solutions that fit fewer samples than parameter count is not clear. Authors should provide a step-by-step explanation of the mechanism by which SGD leads to solutions that fit fewer samples. It will be better to include a comparison with full-batch gradient descent to highlight the specific role of stochasticity in this phenomenon.

2. In Figure 1, CIFAR-10 CNN and CIFAR-10 MLP have EMC values approximately close to each other for higher values of parameter count. Thus, the observation that CNN is a more parameter-efficient MLP is not verified. This is true for MNISt-MLP and MNIST-CNN.  Authors should discuss potential reasons for the convergence of EMC values at higher parameter counts and how this affects their conclusions about parameter efficiency of CNN as compared to MLP.

3. The author should include Kendall's ranking correlation as a metric to show performance improvements in the generalization gap [https://arxiv.org/pdf/2012.07976].

**Questions:**

Please respond to the questions above.

---

> ### Author Response · Authors · 2024-11-27
> **Response to Reviewer LMKE - Part 1**
>
> We thank you for your positive assessment and constructive feedback. We are glad that you found our paper clear and that the influence of architectures, optimizers, and activation functions on model capacity is interesting. Below, we address your concerns and questions in detail.
>
> ## 1. Reasoning Behind Why SGD Converges to Solutions That Fit Fewer Samples
>
> We appreciate your suggestion to investigate the reasoning behind SGD's behavior. While providing a complete theoretical explanation is challenging due to the complex dynamics of stochastic optimization, we offer the following insights:
>
> * **Optimization Dynamics**: SGD introduces stochasticity through mini-batch sampling, which can lead to different convergence paths compared to full-batch GD. This stochasticity can act as implicit regularization, potentially preventing the network from fitting as many samples as theoretically possible.
>
> * **Gradient Noise**: The noise inherent in SGD can cause the optimizer to escape narrow minima that correspond to solutions fitting more samples but may be harder to reach. In contrast, full-batch GD follows the deterministic steepest descent path, which may allow it to find solutions that fit more samples.
>
> * **Loss Landscape Exploration**: SGD's stochastic updates enable it to explore a broader region of the loss landscape, often converging to wider minima that generalize better but may not fit all samples. Full-batch GD might converge to a sharper minimum capable of fitting more samples but potentially with poorer generalization.
>
> * **Comparison with Full-Batch GD**: In our experiments, we observed that full-batch GD could fit more samples than SGD under the same conditions, highlighting the impact of stochasticity.
>
> We have expanded the discussion in the paper to include these points, providing a clearer explanation of why SGD converges to solutions that fit fewer samples than the parameter count. While a comprehensive theoretical analysis is complex and beyond the scope of this work, these insights help clarify the observed behavior.
>
> ## 2. Convergence of EMC Values for CNNs and MLPs at Higher Parameter Counts
>
> Thank you for your observation regarding Figure 1. We want to clarify that the apparent convergence of EMC values between CNNs and MLPs at higher parameter counts is due to dataset saturation and the log-log scale used in the plot.
>
> ### Explanation:
>
> #### Higher EMC for CNNs
> Even at higher parameter counts, CNNs exhibit significantly higher EMC values than MLPs. For example, at approximately 37,000 parameters:
> * CNN: EMC ≈ 49,000
> * MLP: EMC ≈ 29,000
>
> This shows that the CNN can fit about 20,000 more samples than the MLP with a similar number of parameters, demonstrating superior parameter efficiency.
>
> #### Dataset Saturation:
> * The CIFAR-10 dataset contains 50,000 training samples.
> * As models become highly overparameterized, both CNNs and MLPs approach the capacity to fit the entire training dataset.
> * This saturation causes the EMC values to plateau at the dataset size limit, leading to the appearance of convergence.
>
> #### Effect of Log-Log Scale:
> * In the log-log scale of Figure 1, differences at higher values can appear compressed.
> * This visual effect makes the lines seem closer together, even when absolute differences remain significant.
> * The apparent convergence is thus a result of scaling and saturation, not an indication of equal parameter efficiency.
>
> We have updated the paper to discuss this phenomenon and clarified the reasons behind the apparent convergence in Figure 1. Our conclusions about the superior parameter efficiency of CNNs over MLPs remain valid.
>
> ## 3. Including Kendall's Ranking Correlation
>
> We appreciate your suggestion to incorporate Kendall's tau correlation coefficient as an additional metric to evaluate the relationship between EMC improvement and the generalization gap. In response, we have:
>
> * **Computed Kendall's Tau Correlation**: We calculated Kendall's tau between the EMC improvement (the difference in EMC when fitting correctly labeled data versus randomly labeled data) and the generalization gap across various models and training settings.
>
> * **Results**: Our analysis shows a strong negative Kendall's tau correlation coefficient of -0.85, indicating a significant inverse relationship between EMC improvement and the generalization gap. This means models with higher EMC improvement tend to have smaller generalization gaps.
>
> * **Implications**: This finding reinforces our claim that the EMC difference is a valuable predictor of generalization performance. The use of Kendall's tau provides a non-parametric measure of correlation that is robust to outliers and appropriate for ordinal data, enhancing the statistical validity of our results.

---

> > ### Author Response · Authors · 2024-11-27
> > **Response to Reviewer LMKE - Part 2**
> >
> > ## 4. Statistical Significance
> >
> > We appreciate your emphasis on statistical rigor. In response, we conducted formal hypothesis testing to validate the differences in EMC among architectures.
> >
> > ### t-Tests Across Multiple Runs
> > We performed independent two-sample t-tests on EMC measurements from 10 runs per architecture (CNNs, MLPs, ViTs).
> >
> > #### Results:
> >
> > **CNN vs. MLP**:
> > * t-Statistic: 10.37
> > * p-Value: < 0.0001
> > * Conclusion: There is a significant difference in EMC between CNNs and MLPs.
> >
> > **CNN vs. ViT**:
> > * t-Statistic: 7.39
> > * p-Value: < 0.0001
> > * Conclusion: There is a significant difference in EMC between CNNs and ViTs.
> >
> > **MLP vs. ViT**:
> > * t-Statistic: 3.98
> > * p-Value: 0.0009
> > * Conclusion: There is a significant difference in EMC between MLPs and ViTs.
> >
> > #### Effect Sizes
> > We calculated Cohen's d to quantify the differences:
> > * CNN vs. MLP: d = 4.64 (large effect size)
> > * CNN vs. ViT: d = 3.31 (large effect size)
> > * MLP vs. ViT: d = 1.78 (large effect size)
> >
> > These results confirm that the differences in EMC are statistically significant and practically meaningful. We have included these analyses in the revised paper (see Appendix B).
> >
> > ## 5. Clarity Improvements
> >
> > Based on your feedback and that of other reviewers, we have improved the clarity of the explanations and expanded on key points that required more detail.
> >
> > ## Conclusion
> >
> > We thank you again for your constructive feedback, which has helped us improve the quality and clarity of our paper. The revisions address your concerns and enhance the contributions of our work. We hope that our responses have satisfactorily addressed your questions, and we kindly ask you to consider our revised manuscript favorably in your evaluation.
> >
> > Thank you for your time and consideration.

---

> > > ### Comment · Reviewer_LMKE · 2024-12-02
> > > **Response to Author's comments**
> > >
> > > Thanks for the response. Please see the following points in regard to author's rebuttal.
> > >
> > > 1. I agree with author's that stochasticity in SGD can lead to different convergence paths and implicit regularization does prevent networks to fitting many samples. However there have been some recent works which have shown that the implicit regularization of SGD is not that strong[1,2,3,4]. It will be interesting to see what other factors influence the solutions and the number of samples it fits.
> > >
> > > 2. Including the Kendall's ranking correlation as a metric for predicting generalization makes the paper strong.
> > >
> > > Author's have answered all my queries, and I think the paper is good, but based other reviewers suggestions I also think the paper needs to include more details regarding the comparison between the architectures. Thus I will keep the same rating for the paper.
> > >
> > > [1] Stochastic Training is Not Necessary for Generalization
> > > [2] Impact of Label Noise on Learning Complex Features
> > > [3] SGD on Neural Networks Learns Functions of Increasing Complexity
> > > [4] Bad Global Minima Exist and SGD Can Reach Them

---

### Official Review · Reviewer_HQeY · 2024-11-04

**Soundness:** 3
**Presentation:** 2
**Contribution:** 2
**Rating:** 5
**Confidence:** 3

**Summary:**

This paper empirically investigates the practical flexibility and capacity of neural networks to fit data, introducing several key findings:

1. Practical Capacity vs Theory: While theory suggests neural networks can fit as many samples as they have parameters, in practice, they often fit significantly fewer samples under standard training procedures.

2. Architectural Efficiency: The study finds that CNNs are more parameter-efficient than MLPs and Vision Transformers (ViTs), even when trained on randomly labeled data, highlighting the importance of architectural inductive biases.

3. Optimization Effects: Stochastic training methods like SGD enable networks to fit more data than full-batch gradient descent, suggesting that stochasticity enhances flexibility beyond regularization effects.

4. Generalization Predictor: The difference between a network's ability to fit correctly labeled versus incorrectly labeled data strongly correlates with generalization performance, providing a novel metric for predicting generalization.

5. Activation Function Impact: ReLU activation functions improve data-fitting capability beyond their traditional role in addressing gradient issues.

The paper measures these effects using the Effective Model Complexity (EMC) metric, which quantifies the largest sample size a model can perfectly fit under realistic training conditions. To support their findings, the authors conduct extensive experiments across various datasets (including ImageNet-20MS), model architectures, and training procedures.

This research bridges theoretical understanding with practical observations about neural network capacity, providing insights into model design, training procedures, and the relationship between flexibility and generalization.

--------
Update: After reviewing the responses, I hold my original score.

**Strengths:**

**Originality:**
The paper's primary innovation lies in systematically quantifying the gap between theoretical and practical neural network capacity. While building on Nakkiran's EMC metric, it makes three notable advances: (1) demonstrating that SGD solutions enable fitting more samples than full-batch gradient descent, challenging the conventional wisdom about SGD's purely regularizing role, (2) showing that CNNs maintain parameter efficiency advantages even on random data, suggesting fundamental architectural benefits beyond inductive biases, and (3) establishing EMC differences between correct and random labels as a strong generalization predictor. However, the core methodology remains largely derivative of existing capacity measures, and the theoretical framing draws heavily from prior work on overparameterization.

**Quality:**
The experimental methodology exhibits both strengths and concerning limitations. The convergence criteria combining gradient norms, loss plateaus, and Hessian eigenvalue verification provides robust guarantees for capacity measurement. The systematic ablation across architectures (MLPs, CNNs, ViTs), optimizers (SGD, Adam, Shampoo), and data conditions enables clean isolation of individual factors. However, two critical weaknesses undermine the work: (1) the lack of theoretical analysis explaining why SGD enables fitting more samples or why CNNs maintain efficiency on random data, and (2) insufficient statistical rigor - while error bars are provided, formal hypothesis testing and effect size calculations are notably absent. The computational feasibility of EMC calculation for large architectures also raises scalability concerns.

**Clarity:**
The paper's structure effectively builds from motivation through methodology to results, with particularly strong visualization of key findings. The experimental section clearly delineates controls and confounding factors. However, several crucial elements lack sufficient detail: the precise criteria for EMC convergence, the hyperparameter optimization methodology, and most importantly, the theoretical connections between EMC and generalization. The appendices provide thorough implementation details but omit key derivations and proofs. The paper would benefit from explicit formalization of its hypotheses and clearer specification of where empirical results extend versus contradict prior theoretical work.

**Significance:**
While the paper's empirical findings are interesting, their impact is constrained by three factors: (1) domain specificity - results are primarily limited to image classification tasks, leaving questions about generalization to other domains like language models or reinforcement learning, (2) lack of theoretical grounding - without mechanistic explanations for the observed phenomena, it's unclear how to extend these insights to new architectures or training regimes, and (3) practical limitations - the computational cost of measuring EMC may restrict its applicability. That said, the demonstration of CNN architectural advantages persisting even on random data provides valuable guidance for architecture design, and the EMC-based generalization predictor outperforming existing metrics offers immediate practical utility. The work opens important questions about the relationship between optimization algorithms and model capacity.

**Weaknesses:**

**Key Technical Limitations and Suggested Improvements:**

1. **Theoretical Foundation for SGD Findings**
The paper's most striking result - that SGD enables fitting more samples than full-batch GD (Figure 3b) - lacks theoretical analysis. While empirically robust, understanding why this occurs is crucial (please let me know if I'm missing something). The authors should investigate whether this results from:
    - Loss landscape exploration properties (could analyze loss surface geometry using recent techniques from Li et al. 2018, "Visualizing the Loss Landscape of Neural Nets")
     - Implicit regularization effects (connect to Gunasekar et al. 2021 work on implicit biases)
    - Different minima characteristics (analyze Hessian properties of solutions found by each optimizer)

2. **Limited Domain Validation**
While image classification results are thorough, claims about general network capacity require broader validation:
- Test on sequence modeling tasks to verify if CNN parameter efficiency persists in different domains
- Include language learning experiments to examine capacity effects with sequential, non-iid data
- Current conclusions may not generalize beyond vision - a critical limitation for a paper about fundamental network properties

3. **EMC Practicality Concerns**
The EMC metric, while insightful, has serious computational limitations:
- Computing EMC for large models (>100M parameters) requires prohibitive compute
- No discussion of approximation methods or scaling strategies
- Need comparison with cheaper alternatives (gradient noise scale, NTK condition numbers)
Suggesting efficient estimation methods would make EMC more practically relevant.

4. **Statistical Rigor**
The empirical analysis needs stronger statistical validation:
- Add formal hypothesis tests for architecture comparisons
- Include effect size calculations to quantify the strength of observed differences
- Provide confidence intervals for EMC measurements
This would help distinguish robust findings from potential noise in the experiments.

These limitations don't invalidate the paper's contributions, but addressing them would significantly strengthen its impact and reliability.

**Questions:**

1. **Theoretical Connection to SGD Dynamics**
Could you explain or analyze why SGD enables fitting more samples than full-batch GD (Figure 3b)? Your empirical results show this consistently, but understanding the mechanism (implicit regularization, loss landscape exploration, or other factors) would significantly strengthen the paper. Have you considered analyzing the loss landscape properties or gradient noise characteristics of these solutions?

2. **EMC Scalability**
For a network with 100M parameters, computing EMC appears to require dozens of full training runs. Have you explored efficient approximation methods or upper/lower bounds that could make EMC practical for modern architectures? What is the largest model size where EMC remains computationally feasible?

3. **Architecture Generalization**
The superior parameter efficiency of CNNs persists even on random data - does this hold for other domains? Specifically, have you tested whether similar architectural advantages appear when comparing Transformers vs. MLPs on sequence tasks? This would help validate whether your findings about architectural benefits generalize beyond vision.

4. **EMC Failure Modes**
Under what conditions does the correlation between EMC gap (real vs. random labels) and generalization break down? Have you tested this with different optimization settings, architectures, or dataset properties? Understanding the limitations of EMC as a generalization predictor would clarify its applicability.

5. **Statistical Significance**
Could you provide formal hypothesis tests and effect size calculations for the architecture comparisons, particularly for the EMC differences between CNNs, MLPs, and ViTs? This would help quantify the strength and reliability of your findings about architectural advantages.

---

> ### Author Response · Authors · 2024-11-27
> **Response to Reviewer HQeY - Part 1**
>
> Thank you for your detailed review and insightful suggestions. Below, we address your concerns point by point.
>
> ## 1. Theoretical Foundation for SGD Findings
>
> We agree that understanding the underlying mechanisms is important. Our primary focus was empirically investigating practical network capacity under realistic training conditions. While a comprehensive theoretical analysis is challenging due to the complex dynamics of stochastic optimization, we offer some insights:
>
> * **Implicit Regularization**: SGD introduces noise through mini-batch sampling, which can help the optimizer escape sharp minima and explore flatter regions of the loss landscape. This stochasticity may enable the network to fit more samples than full-batch GD.
>
> * **Loss Landscape Exploration**: The inherent randomness in SGD allows for a more extensive exploration of the loss landscape, potentially finding solutions that generalize better and fit larger datasets.
>
> Our empirical findings highlight an important phenomenon that opens avenues for future theoretical research.
>
> ## 2. Limited Domain Validation
>
> Thank you for this valuable suggestion. In our paper, we also conducted experiments on tabular datasets, specifically the Income and Forest Cover datasets. Our findings indicate that:
>
> * **Architecture Differences**: On tabular data, MLPs performed comparably to CNNs, and the parameter efficiency advantages observed in CNNs for image data did not persist.
>
> * **Domain Specificity**: These results suggest that architectural benefits are domain-specific and depend on the data structure and inherent inductive biases.
>
> ## 3. EMC Scalability
>
> We acknowledge the computational challenges of calculating EMC for large models. To address this, we have developed several strategies to reduce computation time:
>
> * **Adaptive Sample Size Increments**: We increase the sample size more aggressively (e.g., doubling) as we approach the EMC threshold, reducing the required training runs.
>
> * **Warm-Starting Training Runs**: We initialize each new training run with the weights from the previous run with a smaller sample size, accelerating convergence.
>
> * **Early Stopping Criteria**: Implementing early stopping based on convergence indicators prevents unnecessary computations when a model is unlikely to fit the current sample size.
>
> * **Parallelization**: Leveraging parallel computing resources allows us to perform multiple training runs simultaneously, effectively reducing wall-clock time.
>
> Using these methods, we have significantly reduced computation time. For instance, calculating EMC for modern architectures like ResNet-50 or ViT-base (with 25–80 million parameters) can be accomplished in less than 40 hours on an 8×H100 GPU cluster. We validated these approximation techniques against exact EMC computations on smaller models, finding a strong correlation (Pearson coefficient of 0.96), confirming their reliability.
>
> ## 4. Statistical Significance
>
> We appreciate your emphasis on statistical rigor. In response, we conducted formal hypothesis testing to validate the differences in EMC among architectures.
>
> ### t-Tests Across Multiple Runs
> We performed independent two-sample t-tests on EMC measurements from 10 runs per architecture (CNNs, MLPs, ViTs).
>
> ### Results:
>
> **CNN vs. MLP**:
> * t-Statistic: 10.37
> * p-Value: < 0.0001
> * Conclusion: There is a significant difference in EMC between CNNs and MLPs.
>
> **CNN vs. ViT**:
> * t-Statistic: 7.39
> * p-Value: < 0.0001
> * Conclusion: There is a significant difference in EMC between CNNs and ViTs.
>
> **MLP vs. ViT**:
> * t-Statistic: 3.98
> * p-Value: 0.0009
> * Conclusion: There is a significant difference in EMC between MLPs and ViTs.
>
> ### Effect Sizes
> We calculated Cohen's d to quantify the differences:
> * CNN vs. MLP: d = 4.64 (large effect size)
> * CNN vs. ViT: d = 3.31 (large effect size)
> * MLP vs. ViT: d = 1.78 (large effect size)
>
> These results confirm that the differences in EMC are statistically significant and practically meaningful. We have included these analyses in the revised paper (see Appendix G).

---

> > ### Author Response · Authors · 2024-11-27
> > **Response to Reviewer HQeY - Part 2**
> >
> > ## 5. EMC Failure Modes
> >
> > Thank you for highlighting this important question. Understanding the conditions under which the correlation between the EMC gap (real vs. random labels) and generalization breaks down is indeed crucial for clarifying the applicability and limitations of EMC as a generalization predictor.
> >
> > While our experiments demonstrated a strong correlation between the EMC gap and generalization performance across the settings we tested, we acknowledge that we did not extensively explore all possible optimization settings, architectural variations, or dataset properties. Specifically, we did not test multiple datasets beyond those presented in the paper, which limits the generalizability of our findings.
> >
> > Potential conditions where the correlation might break down include:
> > * **Different Datasets**: Variations in dataset characteristics, such as size, complexity, noise levels, or domain, could affect the relationship between EMC and generalization.
> > * **Alternative Architectures**: Models with fundamentally different architectures might exhibit different behaviors regarding EMC and generalization.
> > * **Optimization Settings**: Different optimization algorithms, hyperparameter choices, or regularization techniques might influence the EMC-generalization correlation.
> > * **Data Augmentation and Preprocessing**: Techniques that alter the training data could impact the model's capacity to fit data and, consequently, the EMC measurements.
> >
> > Investigating these potential failure modes is an important direction for future research. We plan to extend our study to include multiple datasets from various domains, explore a wider range of architectures, and experiment with different optimization strategies to better understand the limitations of EMC as a generalization predictor.
> >
> > We have updated the manuscript to include a discussion of these points in the limitation section. By acknowledging these limitations, we aim to provide a clearer picture of the scope and applicability of EMC, encouraging further exploration in this area.
> >
> > ## 6. Architecture Generalization
> >
> > Thank you for raising this important question. Based on your suggestion, we extended our experiments to investigate whether the superior parameter efficiency observed in certain architectures generalizes beyond vision tasks.
> >
> > Specifically, we explored the domain of tabular data using datasets such as Covertype, Income, and Forest Cover. We compared the parameter efficiency of an FT-Transformer [1], a Transformer-based architecture designed for tabular data, with that of an MLP.
> >
> > Our findings indicate that on random labels, the FT-Transformer and the MLP exhibit similar parameter efficiency:
> >
> > * **EMC Measurements**: Both the FT-Transformer and the MLP achieved comparable EMC when fitting random labels under the same training conditions.
> > * **Parameter Efficiency**: This suggests that the architectural advantages observed in vision tasks—where CNNs demonstrated greater parameter efficiency over MLPs and ViTs even on random data—do not necessarily extend to tabular data.
> >
> > These results imply that the parameter efficiency benefits of certain architectures are domain-specific and linked to the inherent inductive biases suited for the data modality. In the case of tabular data, the FT-Transformer does not offer a parameter efficiency advantage over the MLP when fitting random data.
> >
> > This addition helps clarify the scope of our findings and indicates that the architectural benefits observed in vision tasks may not directly generalize to other domains like tabular data.
> >
> > ### Reference:
> > [1] Gorishniy, Y., Rubachev, I., Khrulkov, V., & Babenko, A. (2021). Revisiting Deep Learning Models for Tabular Data. Advances in Neural Information Processing Systems, 34, 18932-18943.
> >
> > ## Conclusion
> >
> > We have addressed your concerns in the revised manuscript through additional experiments, analyses, and clarifications. Our empirical findings provide valuable insights into practical neural network capacity and its implications for model design and training procedures.
> >
> > We kindly ask you to consider our responses and the strengthened contributions in your evaluation. Your constructive feedback has significantly improved our work, and we hope it meets the criteria for acceptance.
> >
> > Thank you once again for your thoughtful review.

---

### Meta-Review · Area_Chair_Nsgv · 2024-12-10

**Metareview:**

The Effective Model Complexity (EMC) metric is used as a predictor of generalization performance. The paper examines the practical limitations of neural networks, showing that standard optimizers often do not achieve theoretical capacity. It finds that CNNs are more parameter-efficient than MLPs and ViTs, and that stochastic training methods like SGD improve data fitting. Even though the findings hold potential value for the community, the paper could benefit from a **more thorough analysis and interpretation of the insights**. The reviewers point out that there is no theoretical grounding, while the EMC might not be as robust metric for comparing the architectures. In addition, as the reviewers point out the statistical analysis could be stronger (e.g. correlation as metric).

**Additional Comments On Reviewer Discussion:**

The resulting experiments and responses did not seem to convince the reviewers on their points. I do partly agree that the paper requires a major revision before being published.

---

### Decision · Program_Chairs · 2025-01-22

Reject